# Tight Sample Complexity of Learning One-hidden-layer Convolutional Neural Networks

**Yuan Cao**
Department of Computer Science
University of California, Los Angeles
CA 90095, USA
yuancao@cs.ucla.edu

**Quanquan Gu**
Department of Computer Science
University of California, Los Angeles
CA 90095, USA
qgu@cs.ucla.edu

## Abstract

We study the sample complexity of learning one-hidden-layer convolutional neural networks (CNNs) with non-overlapping filters. We propose a novel algorithm called approximate gradient descent for training CNNs, and show that, with high probability, the proposed algorithm with random initialization grants a linear convergence to the ground-truth parameters up to statistical precision. Compared with existing work, our result applies to general non-trivial, monotonic and Lipschitz continuous activation functions including ReLU, Leaky ReLU, Sigmod and Softplus etc. Moreover, our sample complexity beats existing results in the dependency of the number of hidden nodes and filter size. In fact, our result matches the information-theoretic lower bound for learning one-hidden-layer CNNs with linear activation functions, suggesting that our sample complexity is tight. Our theoretical analysis is backed up by numerical experiments.

## 1 Introduction

Deep learning is one of the key research areas in modern artificial intelligence. Deep neural networks have been successfully applied to various fields including image processing [25], speech recognition [20] and reinforcement learning [33]. Despite the remarkable success in a broad range of applications, theoretical understandings of neural network models remain largely incomplete: the high non-convexity of neural networks makes convergence analysis of learning algorithms very difficult; numerous practically successful choices of the activation function, twists of the training process and variants of the network structure make neural networks even more mysterious.

One of the fundamental problems in learning neural networks is parameter recovery, where we assume the data are generated from a "teacher" network, and the task is to estimate the ground-truth parameters of the teacher network based on the generated data. Recently, a line of research [41, 16, 38] gives parameter recovery guarantees for gradient descent based on the analysis of local convexity and smoothness properties of the square loss function. The results of Zhong et al. [41] and Fu et al. [16] hold for various activation functions except ReLU activation function, while Zhang et al. [38] prove the corresponding result for ReLU. Their results are for fully connected neural networks and their analysis requires accurate knowledge of second-layer parameters. For instance, Fu et al. [16] and Zhang et al. [38] directly assume that the second-layer parameters are known, while Zhong et al. [41] reduce the second-layer parameters to be $\pm 1$'s with the homogeneity assumption, and then exactly recovers them with a tensor initialization algorithm. Moreover, it may not be easy to generalize the local convexity and smoothness argument to other algorithms that are not based on the exact gradient of the loss function. Another line of research [6, 13, 18, 10] focuses on convolutional neural networks with ReLU activation functions. Brutzkus and Globerson [6], Du et al. [13] provide convergence analysis for gradient descent on parameters of both layers, while Goel et al. [18], Du and Goel [10] proposed new algorithms to learn single-hidden-layer CNNs. However, these results

Table 1: Comparison with related work [13, 14, 18, 10]. Note that Du et al. [14] did not study any specific learning algorithms. All sample complexity results are calculated for standard Gaussian inputs and non-overlapping filters. The D. Convotron stands for Double Convotron, which is an algorithm proposed by Du and Goel [10].

|  | Conv. rate | Sample comp. | Act. fun. | Data input | Overlap | Sec. layer |
|---|---|---|---|---|---|---|
| Du et al. [13] | linear | - | ReLU | Gaussian | no | yes |
| Du et al. [14] | - | $\widetilde{O}((k+r)\cdot\epsilon^{-2})$ | Linear | sub-Gaussian | yes | - |
| Convotron [18] | (sub)linear[1] | $\widetilde{O}(k^2 r\cdot\epsilon^{-2})$ | (leaky) ReLU | symmetric | yes | no |
| D. Convotron [10] | sublinear | $\widetilde{O}(\text{poly}(k,r,\epsilon^{-1}))$ | (leaky) ReLU | symmetric | yes | yes |
| **This paper** | linear | $\widetilde{O}((k+r)\cdot\epsilon^{-2})$ | general | Gaussian | no | yes |

heavily rely on the exact calculation of the population gradient for ReLU networks, and do not provide tight sample complexity guarantees.

In this paper, we study the parameter recovery problem for non-overlapping convolutional neural networks. We aim to develop a new convergence analysis framework for neural networks that: (i) works for a class of general activation functions, (ii) does not rely on ad hoc initialization, (iii) can be potentially applied to different variants of the gradient descent algorithm. The main contributions of this paper is as follows:

- We propose an approximate gradient descent algorithm that learns the parameters of both layers in a non-overlapping convolutional neural network. With weak requirements on initialization that can be easily satisfied, the proposed algorithm converges to the ground-truth parameters linearly up to statistical precision.

- Our convergence result holds for all non-trivial, monotonic and Lipschitz continuous activation functions. Compared with the results in Brutzkus and Globerson [6], Du et al. [13], Goel et al. [18], Du and Goel [10], our analysis does not rely on any analytic calculation related to the activation function. We also do not require the activation function to be smooth, which is assumed in the work of Zhong et al. [41] and Fu et al. [16].

- We consider the empirical version of the problem where the estimation of parameters is based on $n$ independent samples. We avoid the usual analysis with sample splitting by proving uniform concentration results. Our method outperforms the state-of-the-art results in terms of sample complexity. In fact, our result for general non-trivial, monotonic and Lipschitz continuous activation functions matches the lower bound given for linear activation functions in Du et al. [14], which implies the statistical optimality of our algorithm.

Detailed comparison between our results and the state-of-the-art on learning one-hidden-layer CNNs is given in Table 1. We compare the convergence rates and sample complexities obtained by recent work with our result. We also summarize the applicable activation functions and data input distributions, and whether overlapping/non-overlapping filters and second layer training are considered in each of the work.

**Notation:** Let $\mathbf{A} = [A_{ij}] \in \mathbb{R}^{d\times d}$ be a matrix and $\mathbf{x} = (x_1,...,x_d)^\top \in \mathbb{R}^d$ be a vector. We use $\|\mathbf{x}\|_q = (\sum_{i=1}^d |x_i|^q)^{1/q}$ to denote $\ell_q$ vector norm for $0 < q < +\infty$. The spectral and Frobenius norms of $\mathbf{A}$ are denoted by $\|\mathbf{A}\|_2$ and $\|\mathbf{A}\|_F$. For a symmetric matrix $\mathbf{A}$, we denote by $\lambda_{\max}(\mathbf{A})$, $\lambda_{\min}(\mathbf{A})$ and $\lambda_i(\mathbf{A})$ the maximum, minimum and $i$-th largest eigenvalues of $\mathbf{A}$. We denote by $\mathbf{A} \succeq 0$ that $\mathbf{A}$ is positive semidefinite (PSD). Given two sequences $\{a_n\}$ and $\{b_n\}$, we write $a_n = O(b_n)$ if there exists a constant $0 < C < +\infty$ such that $a_n \le C\,b_n$, and $a_n = \Omega(b_n)$ if $a_n \le C\,b_n$ for some constant $C$. We use notations $\widetilde{O}(\cdot), \widetilde{\Omega}(\cdot)$ to hide the logarithmic factors. Finally, we denote $a \wedge b = \min\{a,b\}, a \vee b = \max\{a,b\}$.

## 2 Related Work

There has been a vast body of literature on the theory of deep learning. We will review in this section the most relevant work to ours.

It is well known that neural networks have remarkable expressive power due to the universal approximation theorem [22]. However, even learning a one-hidden-layer neural network with a sign activation can be NP-hard [5] in the realizable case. In order to explain the success of deep learning in various applications, additional assumptions on the data generating distribution have been explored such as symmetric distributions [4] and log-concave distributions [24]. More recently, a line of research has focused on Gaussian distributed input for one-hidden-layer or two-layer networks with different structures [23, 35, 6, 27, 41, 40, 17, 38, 16]. Compared with these results, our work aims at providing tighter sample complexity for more general activation functions.

A recent line of work [28, 32, 29, 26, 15, 2, 11, 42, 1, 3, 7] studies the training of neural networks in the over-parameterized regime. Mei et al. [28], Shamir [32], Mei et al. [29] studied the optimization landscape of over-parameterized neural networks. Li and Liang [26], Du et al. [15], Allen-Zhu et al. [2], Du et al. [11], Zou et al. [42] proved that gradient descent can find the global minima of over-parameterized neural networks. Generalization bounds under the same setting are studied in Allen-Zhu et al. [1], Arora et al. [3], Cao and Gu [7]. Compared with these results in the over-parameterized setting, the parameter recovery problem studied in this paper is in the classical setting, and therefore different approaches need to be taken for the theoretical analysis.

This paper studies convolutional neural networks (CNNs). There are not much theoretical literature specifically for CNNs. The expressive power of CNNs is shown in Cohen and Shashua [8]. Nguyen and Hein [30] study the loss landscape in CNNs and Brutzkus and Globerson [6] show the global convergence of gradient descent on one-hidden-layer CNNs. Du et al. [12] extend the result to non-Gaussian input distributions with ReLU activation. Zhang et al. [39] relax the class of CNN filters to a reproducing kernel Hilbert space and prove the generalization error bound for the relaxation. Gunasekar et al. [19] show that there is an implicit bias in gradient descent on training linear CNNs.

## 3 The One-hidden-layer Convolutional Neural Network

In this section we formalize the one-hidden-layer convolutional neural network model. In a convolutional network with neuron number $k$ and filter size $r$, a filter $\mathbf{w} \in \mathbb{R}^r$ interacts with the input $\mathbf{x}$ at $k$ different locations $\mathcal{I}_1, \ldots, \mathcal{I}_k$, where $\mathcal{I}_1, \ldots, \mathcal{I}_k \subseteq \{1, 2, \ldots, d\}$ are index sets of cardinality $r$. Let $\mathcal{I}_j = \{p_{j1}, \ldots, p_{jr}\}, j = 1, \ldots, k$, then the corresponding selection matrices $\mathbf{P}_1, \ldots, \mathbf{P}_k$ are defined as $\mathbf{P}_j = (\mathbf{e}_{p_{j1}}, \ldots, \mathbf{e}_{p_{jr}})^\top, j = 1, \ldots, k$.

We consider a convolutional neural network of the form

$$y = \sum_{j=1}^{k} v_j \sigma(\mathbf{w}^\top \mathbf{P}_j \mathbf{x}_i),$$

where $\sigma(\cdot)$ is the activation function, and $\mathbf{w} \in \mathbb{R}^r$, $\mathbf{v} \in \mathbb{R}^k$ are the first and second layer parameters respectively. Suppose that we have $n$ samples $\{(\mathbf{x}_i, y_i)\}_{i=1}^{n}$, where $\mathbf{x}_1, \ldots, \mathbf{x}_n \in \mathbb{R}^d$ are generated independently from standard Gaussian distribution, and the corresponding output $y_1, \ldots, y_n \in \mathbb{R}$ are generated from the teacher network with true parameters $\mathbf{w}^*$ and $\mathbf{v}^*$ as follows.

$$y_i = \sum_{j=1}^{k} v_j^* \sigma(\mathbf{w}^{*\top} \mathbf{P}_j \mathbf{x}_i) + \epsilon_i,$$

where $k$ is the number of hidden neurons, and $\epsilon_1, \ldots, \epsilon_n$ are independent sub-Gaussian white noises with $\psi_2$ norm $\nu$. Through out this paper, we assume that $\|\mathbf{w}^*\|_2 = 1$.

The choice of activation function $\sigma(\cdot)$ determines the landscape of the neural network. In this paper, we assume that $\sigma(\cdot)$ is a non-trivial, Lipschitz continuous increasing function.

**Assumption 3.1.** $\sigma$ is 1-Lipschitz continuous: $|\sigma(z_1) - \sigma(z_2)| \leq |z_1 - z_2|$ for all $z_1, z_2 \in \mathbb{R}$.

**Assumption 3.2.** $\sigma$ is a non-trivial (not a constant) increasing function.

**Remark 3.3.** Assumptions 3.1 and 3.2 are fairly weak assumptions satisfied by most practically used activation functions including the rectified linear unit (ReLU) function $\sigma(z) = \max(z, 0)$, the

sigmoid function $\sigma(z) = 1/(1 + e^z)$, the hyperbolic tangent function $\sigma(z) = (e^z - e^{-z})/(e^z + e^{-z})$, and the erf function $\sigma(z) = \int_0^z e^{-t^2/2}dt$. Since we do not make any assumptions on the second layer true parameter $\mathbf{v}^*$, our assumptions can be easily relaxed to any non-trivial, $L$-Lipschitz continuous and monotonic functions for arbitrary fixed positive constant $L$.

# 4 Approximate Gradient Descent

## 4.1 Algorithm Description

In this section we present a new algorithm for the estimation of $\mathbf{w}^*$ and $\mathbf{v}^*$.

Let $\mathbf{y} = (y_1, \ldots, y_n)^\top$, $\Sigma(\mathbf{w}) = [\sigma(\mathbf{w}^\top \mathbf{P}_j \mathbf{x}_i)]_{n \times k}$ and $\xi = \mathbb{E}_{z \sim N(0,1)}[\sigma(z)z]$. The algorithm is given in Algorithm 1. We call it approximate gradient descent because it is derived by simply replacing the $\sigma'(\cdot)$ terms in the gradient of the empirical square loss function by the constant $\xi^{-1}$. It is easy to see that under Assumption 3.1 and 3.2, we have $\xi > 0$. Therefore, replacing $\sigma'(\cdot) > 0$ with $\xi^{-1}$ will not drastically change the gradient direction, and gives us an approximate gradient.

Algorithm 1 is also related to, but different from the Convotron algorithm proposed by Goel et al. [18] and the Double Convotron algorithm proposed by Du and Goel [10]. The Approximate Gradient Descent algorithm can be seen a generalized version of the Convotron algorithm, which only considers optimizing over the first layer parameters of the convolutional neural network. Compared to the Double Convotron, Algorithm 1 implements a simpler update rule based on iterative weight normalization for the first layer parameters, and uses a different update rule for the second layer parameters.

---
**Algorithm 1** Approximate Gradient Descent for Non-overlapping CNN

---
**Require:** Training data $\{(\mathbf{x}_i, y_i)\}_{i=1}^n$, number of iterations $T$, step size $\alpha$, initialization $\mathbf{w}^0 \in S^{r-1}$, $\mathbf{v}^0$.
    **for** $t = 0, 1, 2, \ldots, T-1$ **do**
        $\mathbf{g}_w^t = -\frac{1}{n}\sum_{i=1}^n \left[y_i - \sum_{j=1}^k v_j^t \sigma(\mathbf{w}^{t\top}\mathbf{P}_j\mathbf{x}_i)\right] \cdot \sum_{j'=1}^k \xi^{-1} v_{j'}^t \mathbf{P}_{j'}\mathbf{x}_i$
        $\mathbf{g}_v^t = -\frac{1}{n}\mathbf{\Sigma}^\top(\mathbf{w}^t)[\mathbf{y} - \mathbf{\Sigma}(\mathbf{w}^t)\mathbf{v}^t]$
        $\mathbf{u}^{t+1} = \mathbf{w}^t - \alpha\mathbf{g}_w^t$, $\mathbf{w}^{t+1} = \mathbf{u}^{t+1}/\|\mathbf{u}^{t+1}\|_2$, $\mathbf{v}^{t+1} = \mathbf{v}^t - \alpha\mathbf{g}_v^t$
    **end for**
**Ensure:** $\mathbf{w}^T$, $\mathbf{v}^T$

---

## 4.2 Convergence Analysis of Algorithm 1

In this section we give the main convergence result of Algorithm 1. We first introduce some notations. The following quantities are determined purely by the activation function:

$$\kappa := \mathbb{E}_{z \sim N(0,1)}[\sigma(z)], \ \Delta := \text{Var}_{z \sim N(0,1)}[\sigma(z)], \ L := 1 + |\sigma(0)|, \ \Gamma := 1 + |\sigma(0) - \kappa|,$$

$$\phi(\mathbf{w}, \mathbf{w}') := \text{Cov}_{\mathbf{z} \sim N(\mathbf{0}, \mathbf{I})}[\sigma(\mathbf{w}^\top \mathbf{z}), \sigma(\mathbf{w}'^\top \mathbf{z})].$$

The following lemma shows that the function $\phi(\mathbf{w}, \mathbf{w}')$ can in fact be written as a function of $\mathbf{w}^\top\mathbf{w}'$, which we denote as $\psi(\mathbf{w}^\top\mathbf{w}')$. The lemma also reveals that $\psi(\cdot)$ is an increasing function.

**Lemma 4.1.** Under Assumption 3.2, there exists an increasing function $\psi(\tau)$ such that $\psi(\mathbf{w}^\top\mathbf{w}') = \phi(\mathbf{w}, \mathbf{w}')$, and $\Delta \geq \psi(\tau) > 0$ for all $\tau > 0$.

We further define the following quantities.

$$M = \max\left\{\frac{|\kappa|(2L|\mathbf{1}^\top\mathbf{v}^*| + \sqrt{k})}{\Delta + \kappa^2 k}, |\kappa\mathbf{1}^\top(\mathbf{v}^0 - \mathbf{v}^*)|\right\},$$

$$D = \max\left\{\|\mathbf{v}^0 - \mathbf{v}^*\|_2, \sqrt{\frac{4(1 + 4\alpha\Delta)L^2\|\mathbf{v}^*\|_2^2 + 4\alpha\Delta(\kappa^2 M^2 k + 1) + 2}{\Delta^2(1 - 4\alpha\Delta)}}\right\},$$

$$\rho = \min\left\{\frac{1}{2 + \Delta}\psi\left(\frac{1}{2}\mathbf{w}^{*\top}\mathbf{w}^0\right)\|\mathbf{v}^*\|_2^2, \mathbf{v}^{*\top}\mathbf{v}^0\right\}.$$

Let $D_0 = D + \|\mathbf{v}^*\|_2$. Note that in our problem setting the number of filters $k$ can scale with $n$. However, although $k$ is used in the definition of $M$, $D$, $\rho$ and $D_0$, it is not difficult to check that all these quantities can be upper bounded, as is stated in the following lemma.

**Lemma 4.2.** If $\alpha \leq 1/(8\Delta)$, then $M$, $D$, $D_0$ have upper bounds, and $\rho$ has a lower bound, that only depend on the activation function $\sigma(\cdot)$, the ground-truth $(\mathbf{w}^*, \mathbf{v}^*)$ and the initialization $(\mathbf{w}^0, \mathbf{v}^0)$.

We now present our main result, which states that the iterates $\mathbf{w}^t$ and $\mathbf{v}^t$ in Algorithm 1 converge linearly towards $\mathbf{w}^*$ and $\mathbf{v}^*$ respectively up to statistical accuracy.

**Theorem 4.3.** Let $\delta \in (0,1)$, $\gamma_1 = (1 + \alpha\rho)^{-1/2}$, $\gamma_2 = \sqrt{1 - \alpha\Delta + 4\alpha^2\Delta^2}$, and $\gamma_3 = 1 - \alpha(\Delta + \kappa^2 k)$. Suppose that the initialization $(\mathbf{w}^0, \mathbf{v}^0)$ satisfies

$$\mathbf{w}^{*\top}\mathbf{w}^0 > 0, \ \mathbf{v}^{*\top}\mathbf{v}^0 > 0, \ \kappa^2(\mathbf{1}^\top\mathbf{v}^*)\mathbf{1}^\top(\mathbf{v}^0 - \mathbf{v}^*) \leq \rho, \tag{4.1}$$

and the step size $\alpha$ is chosen such that

$$\alpha \leq \frac{1}{2(\Delta + \kappa^2 k)} \wedge \frac{1}{8\Delta} \wedge \frac{\Delta^2}{(24L^2 + 2\Delta^2)\|\mathbf{v}^*\|_2^2 + 2M^2 k + 10} \wedge \frac{1}{2(\|\mathbf{v}^0 - \mathbf{v}^*\|_2^2 + \|\mathbf{v}^*\|_2^2)}.$$

If

$$c_1\sqrt{\frac{(r+k)\log(c_2 nk/\delta)}{n}} \leq 1 \wedge \frac{\rho\sqrt{k}}{|\mathbf{1}^\top\mathbf{v}^*|} \wedge \frac{\Gamma\sqrt{k(r+k)}}{1 + L + \xi} \wedge \frac{(\Gamma + |\kappa|\sqrt{k})\Gamma\sqrt{r+k}}{L^2 + |\kappa| + \Delta + L}$$

$$\wedge \frac{\xi}{D_0(D_0\Gamma + M + \nu)}\left(1 \wedge \frac{\rho}{1 + \alpha\rho}\mathbf{w}^{*\top}\mathbf{w}^0\right)$$

$$\wedge \frac{1}{(\Gamma + \kappa\sqrt{k})(D_0\Gamma + M + \nu)}\left(1 \wedge \frac{\rho}{\|\mathbf{v}^*\|_2}\right) \tag{4.2}$$

for some large enough absolute constants $c_1$ and $c_2$, then there exists absolute constants $C$ and $C'$ such that, with probability at least $1 - \delta$ we have

$$\|\mathbf{w}^t - \mathbf{w}^*\|_2 \leq \gamma_1^t\|\mathbf{w}^0 - \mathbf{w}^*\|_2 + 8\rho^{-1}\gamma_1^{-2}\eta_w, \tag{4.3}$$

$$\|\mathbf{v}^t - \mathbf{v}^*\|_2 \leq R_1 t^{3/2}(\gamma_1 \vee \gamma_2 \vee \gamma_3)^t + (R_2 + R_3|\kappa|\sqrt{k})(\eta_w + \eta_v), \tag{4.4}$$

for all $t = 0, \ldots, T$, where

$$\eta_w = C\xi^{-1}D_0(D_0\Gamma + M + \nu) \cdot \sqrt{\frac{(r+k)\log(120nk/\delta)}{n}}, \tag{4.5}$$

$$\eta_v = C'(\Gamma + \kappa\sqrt{k})(D_0\Gamma + M + \nu) \cdot \sqrt{\frac{(r+k)\log(120nk/\delta)}{n}}, \tag{4.6}$$

and $R_1$, $R_2$, $R_3$ are constants that only depend on the choice of activation function $\sigma(\cdot)$, the ground-truth parameters $(\mathbf{w}^*, \mathbf{v}^*)$ and the initialization $(\mathbf{w}^0, \mathbf{v}^0)$.

Equation (4.2) is an assumption on the sample size $n$. Although this assumption looks complicated, essentially except $k$ and $r$, all quantities in this condition can be treated as constants, and (4.2) can be interpreted as an assumption that $n \geq \widetilde{\Omega}((1 + \kappa\sqrt{k})\sqrt{r+k})$, which is by no means a strong assumption. The second and third lines of (4.2) are to guarantee $\eta_w \leq 1 \wedge [\rho/(1 + \alpha\rho)\mathbf{w}^{*\top}\mathbf{w}^0]$ and $\eta_v \leq 1 \wedge (\rho/\|\mathbf{v}^*\|_2)$ respectively, while the first line is for technical purposes to ensure convergence.

**Remark 4.4.** Theorem 4.3 shows that with initialization satisfying (4.1), Algorithm 1 linearly converges to true parameters up to statistical error. This condition for initialization can be easily satisfied with random initialization. In Section 4.3, we will give a detailed initialization algorithm inspired by a random initialization method proposed in Du et al. [13].

**Remark 4.5.** Compared with the most relevant convergence results in literature given by Du et al. [13], our result is based on optimizing the empirical loss function instead of the population loss function. In particular, when $\kappa = 0$, Theorem 4.3 proves that Algorithm 1 eventually gives estimation of parameters with statistical error of order $O\left(\sqrt{\frac{(r+k)\log(120nk/\delta)}{n}}\right)$. This rate matches the information-theoretic lower bound for one-hidden-layer convolutional neural networks with linear activation functions. Note that our result holds for general activation functions. Matching the

lower bound of the linear case implies the optimality of our algorithm. Compared with two recent results, namely the Convotron algorithm proposed by Goel et al. [18] and the Double Convotron algorithm proposed by Du and Goel [10], which work for ReLU activation and generic symmetric input distributions, our theoretical guarantee for Algorithm 1 gives a tighter sample complexity for more general activation functions, but requires the data inputs to be Gaussian. We remark that if restricting to ReLU activation function, our analysis can be extended to generic symmetric input distributions as well, and can still provide tight sample complexity.

**Remark 4.6.** A recent result by Du et al. [13] discussed a speed-up in convergence when training non-overlapping CNNs with ReLU activation function. This phenomenon also exists in Algorithm 1. To show it, we first note that in Theorem 4.3, the convergence rate of $\mathbf{w}^t$ and $\mathbf{v}^t$ are essentially determined by $\gamma_1 = (1 + \alpha\rho)^{-1/2}$. For appropriately chosen $\alpha$, $\rho$ being too small (i.e. $\mathbf{w}^{*\top}\mathbf{w}^0$ or $\mathbf{v}^{*\top}\mathbf{v}^0$ being too small, by Lemma 4.1) is the only possible reason of slow convergence. Now by the iterative nature of Algorithm 1, for any $T_1 > 0$, we can analyze the convergence behavior after $T_1$ by treating $\mathbf{w}^{T_1}$ and $\mathbf{v}^{T_1}$ as new initialization and applying Theorem 4.3 again. By Theorem 4.3, even if the initialization gives small $\mathbf{w}^{*\top}\mathbf{w}^0$ and $\mathbf{v}^{*\top}\mathbf{v}^0$, after certain number of iterations, $\mathbf{w}^{*\top}\mathbf{w}^t$ and $\mathbf{v}^{*\top}\mathbf{v}^t$ become much larger and therefore the convergence afterwards gets much faster. This phenomenon is comparable with the two-phase convergence result of Du et al. [13]. However, while Du et al. [13] only show the convergence of second layer parameters for phase II of their algorithm, our result shows that linear convergence of Algorithm 1 starts at the first iteration.

## 4.3 Initialization

To complete the theoretical analysis of Algorithm 1, it remains to show that initialization condition (4.1) can be achieved with practical algorithms. The following theorem is inspired by a similar method proposed by Du et al. [13]. It gives a simple random initialization method that satisfy (4.1).

**Theorem 4.7.** Suppose that $\mathbf{w} \in \mathbb{R}^r$ and $\mathbf{v} \in \mathbb{R}^k$ be vectors generated by $\mathbb{P}_w$ and $\mathbb{P}_v$ with support $S^{r-1}$ and $B(\mathbf{0}, k^{-1/2}|\mathbf{1}^\top\mathbf{v}^*|)$ respectively. Then there exists $(\mathbf{w}^0, \mathbf{v}^0) \in \{(\mathbf{w}, \mathbf{v}), (-\mathbf{w}, \mathbf{v}), (\mathbf{w}, -\mathbf{v}), (-\mathbf{w}, -\mathbf{v})\}$ that satisfies (4.1).

**Remark 4.8.** The proof of Theorem 4.7 is fairly straightforward–the vector $\mathbf{v}$ generated by proposed initialization method in fact satisfies that $\kappa^2(\mathbf{1}^\top\mathbf{v}^*)\mathbf{1}^\top(\mathbf{v} - \mathbf{v}^*) \le 0$. Moreover, it is worth noting that for activation functions with $\kappa = \mathbb{E}_{z\sim N(0,1)}[\sigma(z)] = 0$, the initialization condition $\kappa^2(\mathbf{1}^\top\mathbf{v}^*)\mathbf{1}^\top(\mathbf{v}^0 - \mathbf{v}^*) \le \rho$ is automatically satisfied. Therefore for any vector $\mathbf{w} \in S^{r-1}$ and $\mathbf{v} \in \mathbb{R}^k$, one of $(\mathbf{w}, \mathbf{v}), (-\mathbf{w}, \mathbf{v}), (\mathbf{w}, -\mathbf{v}), (-\mathbf{w}, -\mathbf{v})$ satisfies the initialization condition, making initialization for Algorithm 1 extremely easy.

## 5 Proof of the Main Theory

In this section we give the proof of Theorem 4.3. The proof consists of three steps: (i) prove uniform concentration inequalities for approximate gradients, (ii) give recursive upper bounds for $\|\mathbf{w}^t - \mathbf{w}^*\|_2$ and $\|\mathbf{v}^t - \mathbf{v}^*\|_2$, (iii) derive the final convergence result (4.3) and (4.4).

We first analyze how well the approximate gradients $\mathbf{g}_w^t$ and $\mathbf{g}_v^t$ concentrate around their expectations. Instead of using the classic analysis on $\mathbf{g}_w^t$ and $\mathbf{g}_v^t$ conditioning on all previous iterations $\{\mathbf{w}^s, \mathbf{v}^s\}_{s=1}^t$, we consider uniform concentration over a parameter set $\mathcal{W}_0 \times \mathcal{V}_0$ defined as follows.

$$\mathcal{W}_0 := S^{r-1} = \{\mathbf{w} : \|\mathbf{w}\|_2 = 1\}, \; \mathcal{V}_0 := \{\mathbf{v} : \|\mathbf{v} - \mathbf{v}^*\|_2 \le D, \; |\kappa\mathbf{1}^\top(\mathbf{v} - \mathbf{v}^*)| \le M\}.$$

Define

$$\mathbf{g}_w(\mathbf{w}, \mathbf{v}) = -\frac{1}{n}\sum_{i=1}^n \left[ y_i - \sum_{j=1}^k v_j \sigma(\mathbf{w}^\top \mathbf{P}_j \mathbf{x}_i) \right] \cdot \sum_{j'=1}^k \xi^{-1} v_{j'} \mathbf{P}_{j'} \mathbf{x}_i,$$

$$\mathbf{g}_v(\mathbf{w}, \mathbf{v}) = -\frac{1}{n}\mathbf{\Sigma}^\top(\mathbf{w})[\mathbf{y} - \mathbf{\Sigma}(\mathbf{w})\mathbf{v}],$$

$$\overline{\mathbf{g}}_w(\mathbf{w}, \mathbf{v}) = \|\mathbf{v}\|_2^2 \mathbf{w} - (\mathbf{v}^{*\top}\mathbf{v})\mathbf{w}^*,$$

$$\overline{\mathbf{g}}_v(\mathbf{w}, \mathbf{v}) = (\Delta\mathbf{I} + \kappa^2\mathbf{1}\mathbf{1}^\top)\mathbf{v} - [\phi(\mathbf{w}, \mathbf{w}^*)\mathbf{I} + \kappa^2\mathbf{1}\mathbf{1}^\top]\mathbf{v}^*.$$

The following claim follows by direct calculation.

**Claim 5.1.** For any fixed $\mathbf{w}, \mathbf{v}$, it holds that $\mathbb{E}[\mathbf{g}_w(\mathbf{w}, \mathbf{v})] = \overline{\mathbf{g}}_w(\mathbf{w}, \mathbf{v})$ and $\mathbb{E}[\mathbf{g}_v(\mathbf{w}, \mathbf{v})] = \overline{\mathbf{g}}_v(\mathbf{w}, \mathbf{v})$, where the expectation is taken over the randomness of data.

Our goal is to bound $\sup_{(\mathbf{w}, \mathbf{v}) \in \mathcal{W}_0 \times \mathcal{V}_0} \|\mathbf{g}_w(\mathbf{w}, \mathbf{v}) - \overline{\mathbf{g}}_w(\mathbf{w}, \mathbf{v})\|_2$ and $\sup_{(\mathbf{w}, \mathbf{v}) \in \mathcal{W}_0 \times \mathcal{V}_0} \|\mathbf{g}_v(\mathbf{w}, \mathbf{v}) - \overline{\mathbf{g}}_v(\mathbf{w}, \mathbf{v})\|_2$. A key step for proving such uniform bounds is to show thee uniform Lipschitz continuity of $\mathbf{g}_w(\mathbf{w}, \mathbf{v})$ and $\mathbf{g}_v(\mathbf{w}, \mathbf{v})$, which is given in the following lemma.

**Lemma 5.2.** For any $\delta > 0$, if $n \geq (r + k) \log(324/\delta)$, then with probability at least $1 - \delta$, the following inequalities hold uniformly over all $\mathbf{w}, \mathbf{w}' \in \mathcal{W}_0$ and $\mathbf{v}, \mathbf{v}' \in \mathcal{V}_0$:

$$\|\mathbf{g}_w(\mathbf{w}, \mathbf{v}) - \mathbf{g}_w(\mathbf{w}', \mathbf{v})\|_2 \leq C\xi^{-1} D_0^2 \sqrt{k} \cdot \|\mathbf{w} - \mathbf{w}'\|_2, \tag{5.1}$$

$$\|\mathbf{g}_w(\mathbf{w}, \mathbf{v}) - \mathbf{g}_w(\mathbf{w}, \mathbf{v}')\|_2 \leq C\xi^{-1}(\nu + D_0 L\sqrt{k}) \cdot \|\mathbf{v} - \mathbf{v}'\|_2, \tag{5.2}$$

$$\|\mathbf{g}_v(\mathbf{w}, \mathbf{v}) - \mathbf{g}_v(\mathbf{w}', \mathbf{v})\|_2 \leq C(\nu + D_0 L\sqrt{k})\sqrt{k} \cdot \|\mathbf{w} - \mathbf{w}'\|_2, \tag{5.3}$$

$$\|\mathbf{g}_v(\mathbf{w}, \mathbf{v}) - \mathbf{g}_v(\mathbf{w}, \mathbf{v}')\|_2 \leq CL^2 k \cdot \|\mathbf{v} - \mathbf{v}'\|_2, \tag{5.4}$$

where $C$ is an absolute constant.

If $\mathbf{g}_w$ and $\mathbf{g}_v$ are gradients of some objective function $f$, then Lemma 5.2 essentially proves the uniform smoothness of $f$. However, in our algorithm, $\mathbf{g}_w$ is not the exact gradient, and therefore the results are stated in the form of Lipschitz continuity of $\mathbf{g}_w$. Lemma 5.2 enables us to use a covering number argument together with point-wise concentration inequalities to prove uniform concentration, which is given as Lemma 5.3.

**Lemma 5.3.** Assume that $n \geq (r + k) \log(972/\delta)$, and

$$\xi^{-1} D_0(D_0\Gamma + M + \nu)\sqrt{n(r + k) \log(90nk/\delta)} \geq D_0(D_0 + 1) \vee \xi^{-1}(\nu + D_0^2 + D_0 L),$$

$$(\Gamma + \kappa\sqrt{k})[D_0\Gamma + M + \nu]\sqrt{n(r + k) \log(90nk/\delta)} \geq (\Delta + \kappa + D_0 L) \vee (\nu + D_0 L + L^2).$$

Then with probability at least $1 - \delta$ we have

$$\sup_{(\mathbf{w}, \mathbf{v}) \in \mathcal{W}_0 \times \mathcal{V}_0} \|\mathbf{g}_w(\mathbf{w}, \mathbf{v}) - \overline{\mathbf{g}}_w(\mathbf{w}, \mathbf{v})\|_2 \leq \eta_w, \tag{5.5}$$

$$\sup_{(\mathbf{w}, \mathbf{v}) \in \mathcal{W}_0 \times \mathcal{V}_0} \|\mathbf{g}_v(\mathbf{w}, \mathbf{v}) - \overline{\mathbf{g}}_v(\mathbf{w}, \mathbf{v})\|_2 \leq \eta_v, \tag{5.6}$$

where $\eta_w$ and $\eta_v$ are defined in (4.5) and (4.6) respectively with large enough constants $C$ and $C'$.

We now proceed to study the recursive properties of $\{\mathbf{w}^t\}$ and $\{\mathbf{v}^t\}$. Define

$$\mathcal{W} := \{\mathbf{w} : \|\mathbf{w}\|_2 = 1, \ \mathbf{w}^{*\top}\mathbf{w} \geq \mathbf{w}^{*\top}\mathbf{w}^0/2\},$$

$$\mathcal{V} := \{\mathbf{v} : \|\mathbf{v} - \mathbf{v}^*\|_2 \leq D, \ |\kappa\mathbf{1}^\top(\mathbf{v} - \mathbf{v}^*)| \leq M, \mathbf{v}^{*\top}\mathbf{v} \geq \rho, \ \kappa^2(\mathbf{1}^\top\mathbf{v}^*)\mathbf{1}^\top(\mathbf{v} - \mathbf{v}^*) \leq \rho\}.$$

Then clearly $\mathcal{W} \times \mathcal{V} \subseteq \mathcal{W}_0 \times \mathcal{V}_0$, and therefore the results of Lemma 5.3 hold for $(\mathbf{w}, \mathbf{v}) \in \mathcal{W} \times \mathcal{V}$.

**Lemma 5.4.** Suppose that (5.5) and (5.6) hold. Under the assumptions of Theorem 4.3, in Algorithm 1, if $(\mathbf{w}^t, \mathbf{v}^t) \in \mathcal{W} \times \mathcal{V}$, then

$$\|\mathbf{w}^{t+1} - \mathbf{w}^*\|_2 - 8\rho^{-1}(1 + \alpha\rho)\eta_w \leq \frac{1}{\sqrt{1 + \alpha\rho}}[\|\mathbf{w}^t - \mathbf{w}^*\|_2 - 8\rho^{-1}(1 + \alpha\rho)\eta_w], \tag{5.7}$$

$$|\mathbf{1}^\top(\mathbf{v}^{t+1} - \mathbf{v}^*)| \leq [1 - \alpha(\Delta + \kappa^2 k)]|\mathbf{1}^\top(\mathbf{v}^t - \mathbf{v}^*)| + \alpha L\|\mathbf{w}^t - \mathbf{w}^*\|_2|\mathbf{1}^\top\mathbf{v}^*| + \alpha\sqrt{k}\eta_v, \tag{5.8}$$

$$\|\mathbf{v}^{t+1} - \mathbf{v}^*\|_2^2 \leq (1 - \alpha\Delta + 4\alpha^2\Delta^2)\|\mathbf{v}^t - \mathbf{v}^*\|_2^2 + \left(\frac{\alpha L^2}{\Delta} + 4\alpha^2 L^2\right)\|\mathbf{v}^*\|_2^2\|\mathbf{w}^t - \mathbf{w}^*\|_2^2$$

$$+ 4\alpha^2\kappa^4 k[\mathbf{1}^\top(\mathbf{v}^t - \mathbf{v}^*)]^2 + \left(\frac{2\alpha}{\Delta} + 4\alpha^2\right)\eta_v^2, \tag{5.9}$$

$$(\mathbf{w}^{t+1}, \mathbf{v}^{t+1}) \in \mathcal{W} \times \mathcal{V}. \tag{5.10}$$

It is not difficult to see that the results of Lemma 5.4 imply convergence of $\mathbf{w}^t$ and $\mathbf{v}^t$ up to statistical error. To obtain the final convergence result of Theorem 4.3, it suffices to rewrite the recursive bounds into explicit bounds, which is mainly tedious calculation. We therefore summarize the result as the following lemma, and defer the detailed calculation to appendix.

**Lemma 5.5.** Suppose that (5.7), (5.8) and (5.9) hold for all $t = 0, \ldots, T$. Then

$$\|\mathbf{w}^t - \mathbf{w}^*\|_2 \leq \gamma_1^t \|\mathbf{w}^0 - \mathbf{w}^*\|_2 + 8\rho^{-1}\gamma_1^{-2}\eta_w,$$
$$\|\mathbf{v}^t - \mathbf{v}^*\|_2 \leq R_1 t^3 (\gamma_1 \vee \gamma_2 \vee \gamma_3)^t + (R_2 + R_3|\kappa|\sqrt{k})(\eta_w + \eta_v)$$

for all $t = 0, \ldots, T$, where $\gamma_1 = (1 + \alpha\rho)^{-1/2}$, $\gamma_2 = \sqrt{1 - \alpha\Delta + 4\alpha^2\Delta^2}$, and $\gamma_3 = 1 - \alpha(\Delta + \kappa^2 k)$, and $R_1$, $R_2$, $R_3$ are constants that only depend on the choice of activation function $\sigma(\cdot)$, the ground-truth parameters $(\mathbf{w}^*, \mathbf{v}^*)$ and the initialization $(\mathbf{w}^0, \mathbf{v}^0)$.

We are now ready to present the final proof of Theorem 4.3, which is a straightforward combination of the results of Lemma 5.4 and Lemma 5.5.

*Proof of Theorem 4.3.* By Lemma 5.4, as long as $(\mathbf{w}^0, \mathbf{v}^0) \in \mathcal{W} \times \mathcal{V}$, (5.7), (5.8) and (5.9) hold for all $t = 0, \ldots, T$. Therefore by Lemma 5.5, we have

$$\|\mathbf{w}^t - \mathbf{w}^*\|_2 \leq \gamma_1^t \|\mathbf{w}^0 - \mathbf{w}^*\|_2 + 8\rho^{-1}\gamma_1^{-2}\eta_w,$$
$$\|\mathbf{v}^t - \mathbf{v}^*\|_2 \leq R_1 t^3 (\gamma_1 \vee \gamma_2 \vee \gamma_3)^t + (R_2 + R_3|\kappa|\sqrt{k})(\eta_w + \eta_v)$$

for all $t = 0, \ldots, T$. This completes the proof of Theorem 4.3. $\square$

# 6 Experiments

We perform numerical experiments to backup our theoretical analysis. We test Algorithm 1 together with the initialization method given in Theorem 4.7 for ReLU, sigmoid and hyperbolic tangent networks, and compare its performance with the Double Convotron algorithm proposed by Du and Goel [10]. To give a reasonable comparison, we use a batch version of Double Convotron without the additional noises on unit sphere, which gives the best performance for Double Convotron, and makes it directly comparable with our algorithm. The detailed parameter choices are given as follows:

- For all experiments, we set the number of iterations $T = 100$, sample size $n = 1000$.
- We tune the step size $\alpha$ to maximize performance. Specifically, we set $\alpha = 0.04$ for ReLU, $\alpha = 0.25$ for sigmoid, and $\alpha = 0.1$ for hyperbolic tangent networks. Note that for sigmoid and hyperbolic tangent networks, an inappropriate step size can easily lead to blown up errors for Double Convotron.
- We uniformly generate $\mathbf{w}^*$ from unit sphere, and generate $\mathbf{v}^*$ as a standard Gaussian vector.
- We consider two settings: (i) $k = 15$, $r = 5$, $\widetilde{\nu} = 0.08$, (ii) $k = 30$, $r = 9$, $\widetilde{\nu} = 0.04$, where $\widetilde{\nu}$ is the standard deviation of white Gaussian noises.

The random initialization is performed as follows: we generate $\mathbf{w}$ uniformly over the unit sphere. We then generate a standard Gaussian vector $\mathbf{v}$. If $\|\mathbf{v}\|_2 \geq k^{-1/2}|\mathbf{1}^\top\mathbf{v}^*|/2$, then $\mathbf{v}$ is projected onto the ball $\mathcal{B}(0, k^{-1/2}|\mathbf{1}^\top\mathbf{v}^*|/2)$. We then run the approximate gradient descent algorithm and Double Convotron algorithm starting with each of $(\mathbf{w}, \mathbf{v}), (-\mathbf{w}, \mathbf{v}), (\mathbf{w}, -\mathbf{v}), (-\mathbf{w}, -\mathbf{v})$, and present the results corresponding to the starting point that gives the smallest $\|\mathbf{w}^T - \mathbf{w}^*\|_2$.

Figure 1 gives the experimental results in semi-log plots. We summarize the results as follows.

1. For all the six cases, the approximate gradient descent algorithm eventually reaches a stable state of linear convergence, until reaching very small error.

2. For ReLU networks, both algorithms converges. The convergence of approximated gradient descent algorithm is slower compared with Double Convotron, but it eventually reaches smaller statistical error, indicating a better sample complexity.

3. For sigmoid and hyperbolic tangent networks, not surprisingly, Double Convotron does not converge. In contrast, approximated gradient descent still converges in a linear rate.

The experimental results discussed above clearly demonstrates the validity of our theoretical analysis. In Appendix F, we also present some additional experiments on non-Gaussian inputs, and demonstrate that although this setting is not the focus of our theoretical results, approximate gradient descent still has promising performance on symmetric data distributions.

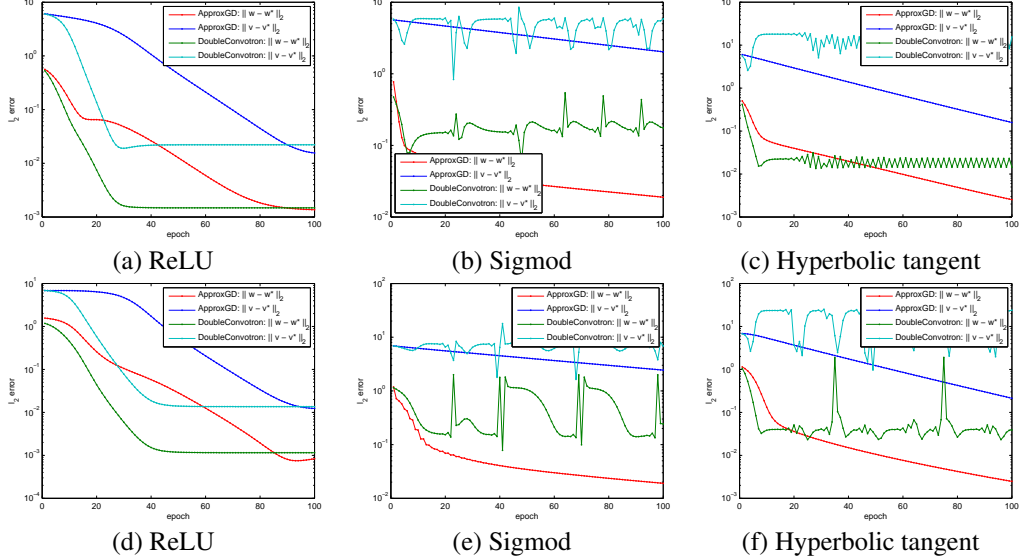

Figure 1: Numerical simulation for Algorithm 1 and the Double Convotron algorithm proposed by Du and Goel [10] with different activation functions, number of hidden nodes and filter sizes. The results for the case $k = 15$, $r = 5$ and $k = 30$, $r = 9$ are shown in (a)-(c) and (d)-(f) respectively. (a), (d) show the results for ReLU networks; (b), (e) give the results for sigmoid networks; and finally the results for hyperbolic tangent activation function are in (c) and (f). All plots are semi-log plots.

## 7 Conclusions and Future Work

We propose a new algorithm namely approximate gradient descent for training CNNs, and show that, with high probability, the proposed algorithm with random initialization can recover the ground-truth parameters up to statistical precision at a linear convergence rate . Compared with previous results, our result applies to a class of monotonic and Lipschitz continuous activation functions including ReLU, Leaky ReLU, Sigmod and Softplus etc. Moreover, our algorithm achieves better sample complexity in the dependency of the number of hidden nodes and filter size. In particular, our result matches the information-theoretic lower bound for learning one-hidden-layer CNNs with linear activation functions, suggesting that our sample complexity is tight. Numerical experiments on synthetic data corroborate our theory. Our algorithms and theory can be extended to learn one-hidden-layer CNNs with overlapping filters. We leave it as a future work. It is also of great importance to extend the current result to deeper CNNs with multiple convolution filters.

## Acknowledgement

We thank the anonymous reviewers and area chair for their helpful comments. This research was sponsored in part by the National Science Foundation CAREER Award IIS-1906169, IIS-1903202, and Salesforce Deep Learning Research Award. The views and conclusions contained in this paper are those of the authors and should not be interpreted as representing any funding agencies.

## Footnotes

[1][18] provided a general sublinear convergence result as well as a linear convergence rate for the noiseless case. We only list their sample complexity result of the noisy case in the table for proper comparison.

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
