[Supplementary Material]

# A Proofs of Lemmas in Section 4

## A.1 Proof of Lemma 4.1

We first present the following lemma. The proof is given in Appendix C.

**Lemma A.1.** Let $f(z)$ and $g(z)$ be two non-trivial increasing functions. Let $Z_1$ and $Z_2$ be zero-mean jointly Gaussian random variables. If $\mathrm{Var}(Z_1) = \mathrm{Var}(Z_2) = 1$ and $\theta = \mathrm{Cov}(Z_1, Z_2) > 0$, then $\mathrm{Cov}[f(Z_1), g(Z_2)]$ is an increasing function of $\theta$, and we have $\mathrm{Cov}[f(z_1), g(z_2)] > 0$.

*Proof of Lemma 4.1.* Since $\mathrm{Cov}(\mathbf{w}^\top \mathbf{Z}, \mathbf{w}'^\top \mathbf{Z}) = \mathbf{w}^\top \mathbf{w}'$, by Lemma A.1 we know there exists an increasing function $\psi(\tau)$ such that $\psi(\mathbf{w}^\top \mathbf{w}') = \phi(\mathbf{w}, \mathbf{w}')$ and $\psi(\tau) > 0$ for $\tau > 0$. $\psi(\tau) \le \Delta$ follows directly by Cauchy-Schwarz inequality. $\qquad\square$

## A.2 Proof of Lemma 4.2

*Proof of Lemma 4.2.* For $M$, if $\kappa = 0$ it is obvious that $M = 0$. If $\kappa \ne 0$, we have

$$\frac{|\kappa|(2L|\mathbf{1}^\top \mathbf{v}^*| + \sqrt{k})}{\Delta + \kappa^2 k} \le \frac{|\kappa|(2L|\mathbf{1}^\top \mathbf{v}^*| + \sqrt{k})}{\kappa^2 k} \le \frac{(2L|\mathbf{1}^\top \mathbf{v}^*| + 1)}{|\kappa|}.$$

This upper bound if $M$ does not depend on $k$.

For $D$, since we assume that $\alpha \le 1/(8\Delta)$, it suffices to show that $\kappa^2 M^2 k$ is bounded. Similar to the bound of $M$, if $\kappa = 0$ clearly $\kappa^2 M^2 k$. If $\kappa \ne 0$, we have

$$\kappa^2 k \cdot \left[\frac{|\kappa|(2L|\mathbf{1}^\top \mathbf{v}^*| + \sqrt{k})}{\Delta + \kappa^2 k}\right]^2 \le \kappa^2 k \cdot \left[\frac{|\kappa|\sqrt{k}(2L|\mathbf{1}^\top \mathbf{v}^*| + 1)}{\kappa^2 k}\right]^2 = (2L|\mathbf{1}^\top \mathbf{v}^*| + 1)^2.$$

Therefore $D$ has an upper bound that only depends on the choice of activation function $\sigma(\cdot)$, the ground-truth parameters $(\mathbf{w}^*, \mathbf{v}^*)$ and the initialization $(\mathbf{w}^0, \mathbf{v}^0)$. The results for $D_0$ and $\rho$ are obvious. $\qquad\square$

# B Proofs of Results in Section 5

In this section we give the proofs of the claims and lemmas used in Section 5.

## B.1 Proof of Claim 5.1

*Proof of Claim 5.1.* Note that for any $i = 1, \ldots, n$, $\mathbf{P}_j \mathbf{x}_i$, $j = 1, \ldots, k$ are independent standard Gaussian random vectors. Therefore we have $v_j \sigma(\mathbf{w}^\top \mathbf{P}_j \mathbf{x}_i) \cdot \xi^{-1} v_{j'} \mathbf{P}_{j'} \mathbf{x}_i = 0$ for $j' \ne j$. Moreover, suppose that $\mathbf{z}$ is a standard Gaussian random vector. Let $\widetilde{\mathbf{w}}_1, \ldots, \widetilde{\mathbf{w}}_{r-1}$ be a set of orthonormal vectors orthogonal to $\mathbf{w}$, then we have

$$\mathbb{E}_{\mathbf{z}}[\sigma(\mathbf{w}^\top \mathbf{z})\mathbf{z}] = \mathbb{E}_{\mathbf{z}}[\sigma(\mathbf{w}^\top \mathbf{z}) \cdot (\mathbf{w}^\top \mathbf{z})] \cdot \mathbf{w} + \sum_{j=1}^{r-1} \mathbb{E}_{\mathbf{z}}[\sigma(\mathbf{w}^\top \mathbf{z}) \cdot (\mathbf{w}_j^\top \mathbf{z})] \cdot \mathbf{w}_j$$

$$= \mathbb{E}_{\mathbf{z}}[\sigma(\mathbf{w}^\top \mathbf{z}) \cdot (\mathbf{w}^\top \mathbf{z})] \cdot \mathbf{w}$$

$$= \xi \cdot \mathbf{w},$$

where the second equality follows by the fact that $\mathbf{w}_j^\top \mathbf{z}$, $j = 1, \ldots, r-1$ are independent of $\mathbf{w}^\top \mathbf{z}$ and have mean $0$. Note that this argument for $\mathbf{w}$ also works for $\mathbf{w}^*$. Therefore, we have

$$\mathbb{E}[\mathbf{g}_w(\mathbf{w}, \mathbf{v})] = -\frac{1}{n}\sum_{i=1}^{n} \mathbb{E}\left\{\left[y_i - \sum_{j=1}^{k} v_j \sigma(\mathbf{w}^\top \mathbf{P}_j \mathbf{x}_i)\right] \cdot \sum_{j'=1}^{k} \xi^{-1} v_{j'} \mathbf{P}_{j'} \mathbf{x}_i\right\}$$

$$= -\frac{1}{n}\sum_{i=1}^{n} \mathbb{E}\left\{\left[\sum_{j=1}^{k} v_j^* \sigma(\mathbf{w}^{*\top} \mathbf{P}_j \mathbf{x}_i) - \sum_{j=1}^{k} v_j \sigma(\mathbf{w}^\top \mathbf{P}_j \mathbf{x}_i)\right] \cdot \sum_{j'=1}^{k} \xi^{-1} v_{j'} \mathbf{P}_{j'} \mathbf{x}_i\right\}$$

$$= -\left(\sum_{j=1}^{k} v_j^* v_j \mathbf{w}^* - \sum_{j=1}^{k} v_j^2 \mathbf{w}\right)$$

$$= \overline{\mathbf{g}}_w(\mathbf{w}, \mathbf{v}).$$

This proves the first result. The second identity $\mathbb{E}[\mathbf{g}_v(\mathbf{w}, \mathbf{v})] = \overline{\mathbf{g}}_v(\mathbf{w}, \mathbf{v})$ directly follows by the definition. $\qquad\square$

## B.2 Proof of Lemma 5.2

*Proof of Lemma 5.2.* Define

$$H_{ww} := \sup_{\substack{\mathbf{w}, \mathbf{w}' \in \mathcal{W}_0 \\ \mathbf{v} \in \mathcal{V}_0}} \frac{\|\mathbf{g}_w(\mathbf{w}, \mathbf{v}) - \mathbf{g}_w(\mathbf{w}', \mathbf{v})\|_2}{\|\mathbf{w} - \mathbf{w}'\|_2}, \quad H_{wv} := \sup_{\substack{\mathbf{v}, \mathbf{v}' \in \mathcal{V}_0 \\ \mathbf{w} \in \mathcal{W}_0}} \frac{\|\mathbf{g}_w(\mathbf{w}, \mathbf{v}) - \mathbf{g}_w(\mathbf{w}, \mathbf{v}')\|_2}{\|\mathbf{v} - \mathbf{v}'\|_2},$$

$$H_{vw} := \sup_{\substack{\mathbf{w}, \mathbf{w}' \in \mathcal{W}_0 \\ \mathbf{v} \in \mathcal{V}_0}} \frac{\|\mathbf{g}_v(\mathbf{w}, \mathbf{v}) - \mathbf{g}_v(\mathbf{w}', \mathbf{v})\|_2}{\|\mathbf{w} - \mathbf{w}'\|_2}, \quad H_{vv} := \sup_{\substack{\mathbf{v}, \mathbf{v}' \in \mathcal{V}_0 \\ \mathbf{w} \in \mathcal{W}_0}} \frac{\|\mathbf{g}_v(\mathbf{w}, \mathbf{v}) - \mathbf{g}_v(\mathbf{w}, \mathbf{v}')\|_2}{\|\mathbf{v} - \mathbf{v}'\|_2}.$$

For any $\delta_1, \ldots, \delta_5 > 0$, we first give the following lemmas.

**Lemma B.1.** *If $n \geq (r + k) \log(34/\delta_1)$, then with probability at least $1 - \delta_1$, we have*

$$\sup_{\substack{\mathbf{a}, \mathbf{a}' \in S^{r-1} \\ \mathbf{b}, \mathbf{b}' \in S^{k-1}}} \frac{1}{n} \sum_{i=1}^{n} \sum_{j=1}^{k} |b_j \mathbf{a}^\top \mathbf{P}_j \mathbf{x}_i| \cdot \left| \sum_{j'=1}^{k} b'_{j'} \mathbf{a}'^\top \mathbf{P}_{j'} \mathbf{x}_i \right| \leq C\sqrt{k},$$

*where $C$ is an absolute constant.*

**Lemma B.2.** *If $n \geq (r + k) \log(68/\delta_2)$, then with probability at least $1 - \delta_2$, we have*

$$\sup_{\substack{\mathbf{a}, \mathbf{a}' \in S^{r-1} \\ \mathbf{b}, \mathbf{b}' \in S^{k-1}}} \frac{1}{n} \sum_{i=1}^{n} \sum_{j=1}^{k} b_j \sigma(\mathbf{a}^\top \mathbf{P}_j \mathbf{x}_i) \cdot \sum_{j'=1}^{k} b'_{j'} \mathbf{a}'^\top \mathbf{P}_{j'} \mathbf{x}_i \leq CL\sqrt{k},$$

*where $C$ is an absolute constant.*

**Lemma B.3.** *If $n \geq (r + k) \log(34/\delta_3)$, then with probability at least $1 - \delta_3$, we have*

$$\sup_{\substack{\mathbf{a}, \mathbf{a}' \in S^{r-1} \\ \mathbf{b}, \mathbf{b}' \in S^{k-1}}} \frac{1}{n} \sum_{i=1}^{n} \sum_{j=1}^{k} |b_j \mathbf{a}^\top \mathbf{P}_j \mathbf{x}_i| \cdot \sum_{j'=1}^{k} |b'_{j'} \mathbf{a}'^\top \mathbf{P}_{j'} \mathbf{x}_i| \leq Ck,$$

*where $C$ is an absolute constant.*

**Lemma B.4.** *If $n \geq (r + k) \log(68/\delta_4)$, then with probability at least $1 - \delta_4$, we have*

$$\sup_{\substack{\mathbf{a}, \mathbf{a}' \in S^{r-1} \\ \mathbf{b}, \mathbf{b}' \in S^{k-1}}} \frac{1}{n} \sum_{i=1}^{n} \sum_{j=1}^{k} |b_j \mathbf{a}^\top \mathbf{P}_j \mathbf{x}_i| \cdot \sum_{j'=1}^{k} |b'_{j'} \sigma(\mathbf{a}'^\top \mathbf{P}_{j'} \mathbf{x}_i)| \leq CLk,$$

*where $C$ is an absolute constant.*

**Lemma B.5.** *If $n \geq (r + k) \log(102/\delta_5)$, then with probability at least $1 - \delta_5$, we have*

$$\sup_{\substack{\mathbf{a}, \mathbf{a}' \in S^{r-1} \\ \mathbf{b}, \mathbf{b}' \in S^{k-1}}} \frac{1}{n} \sum_{i=1}^{n} \sum_{j=1}^{k} b_j \sigma(\mathbf{a}^\top \mathbf{P}_j \mathbf{x}_i) \cdot \sum_{j'=1}^{k} b'_{j'} \sigma(\mathbf{a}'^\top \mathbf{P}_{j'} \mathbf{x}_i) \leq CL^2 k,$$

*where $C$ is an absolute constant.*

**Lemma B.6.** *If $n \geq (r + k) \log(18/\delta_6)$, then with probability at least $1 - \delta_6$, we have*

$$\sup_{\substack{\mathbf{a} \in S^{r-1} \\ \mathbf{b} \in S^{k-1}}} \frac{1}{n} \sum_{i=1}^{n} \epsilon_i \cdot \sum_{j=1}^{k} b_j \mathbf{a}^\top \mathbf{P}_j \mathbf{x}_i \leq C\nu,$$

*where $C$ is an absolute constant.*

**Lemma B.7.** If $n \geq (r + k) \log(18/\delta_7)$, then with probability at least $1 - \delta_7$, we have

$$\sup_{\substack{\mathbf{a} \in S^{r-1} \\ \mathbf{b} \in S^{k-1}}} \frac{1}{n} \sum_{i=1}^{n} |\epsilon_i| \cdot \sum_{j=1}^{k} |b_j \mathbf{a}^\top \mathbf{P}_j \mathbf{x}_i| \leq C\nu\sqrt{k},$$

where $C$ is an absolute constant.

Let $\delta_1 = \delta/9$, $\delta_2 = \delta_4 = 2\delta/9$, $\delta_5 = \delta/3$ and $\delta_6 = \delta_7 = \delta/18$. Then we have $\delta_1 + \delta_2 + \delta_4 + \delta_5 + \delta_6 + \delta_7 = \delta$. By union bound and the assumption that $n \geq (r + k) \log(324/\delta)$, with probability at least $1 - \delta$, the results of Lemmas B.1, B.2, B.4, B.5, B.6, and B.7 all hold. We are now ready to prove (5.1)-(5.4).

**Proof of** (5.1). By Assumption 3.1 and Lemma B.1 we have

$$
\begin{aligned}
H_{ww} &= \sup_{\mathbf{a} \in S^{r-1}, \mathbf{v} \in \mathcal{V}_0} \frac{|\mathbf{a}^\top [\mathbf{g}_w(\mathbf{w}, \mathbf{v}) - \mathbf{g}_w(\mathbf{w}', \mathbf{v})]|}{\|\mathbf{w} - \mathbf{w}^*\|_2} \\
&\leq \sup_{\mathbf{a} \in S^{r-1}, \mathbf{v} \in \mathcal{V}_0} \xi^{-1}\|\mathbf{v}\|_2^2 \cdot \frac{1}{n} \sum_{i=1}^{n} \sum_{j=1}^{k} \left| \frac{v_j}{\|\mathbf{v}\|_2} \frac{(\mathbf{w} - \mathbf{w}')^\top}{\|\mathbf{w} - \mathbf{w}'\|_2} \mathbf{P}_j \mathbf{x}_i \right| \cdot \left| \sum_{j'=1}^{k} \frac{v_{j'}}{\|\mathbf{v}\|_2} \mathbf{a}^\top \mathbf{P}_j \mathbf{x}_i \right| \\
&\leq C_1 \xi^{-1} D_0^2 \sqrt{k},
\end{aligned}
$$

where $C_1$ is an absolute constant.

**Proof of** (5.2) For any $\mathbf{a} \in S^{r-1}$, by definition we have

$$\mathbf{a}^\top [\mathbf{g}_w(\mathbf{w}, \mathbf{v}) - \mathbf{g}_w(\mathbf{w}, \mathbf{v}')] = I_1 + I_2 + I_3 + I_4,$$

where

$$I_1 = -\frac{1}{n} \sum_{i=1}^{n} \sum_{j=1}^{k} v_j^* \sigma(\mathbf{w}^{*\top} \mathbf{P}_j \mathbf{x}_i) \sum_{j'=1}^{k} \xi^{-1}(v_{j'} - v_{j'}')\mathbf{a}^\top \mathbf{P}_{j'} \mathbf{x}_i,$$

$$I_2 = \frac{1}{n} \sum_{i=1}^{n} \sum_{j=1}^{k} (v_j - v_j')\sigma(\mathbf{w}^\top \mathbf{P}_j \mathbf{x}_i) \cdot \sum_{j'=1}^{k} \xi^{-1} v_{j'} \mathbf{a}^\top \mathbf{P}_{j'} \mathbf{x}_i,$$

$$I_3 = \frac{1}{n} \sum_{i=1}^{n} \sum_{j=1}^{k} v_j' \sigma(\mathbf{w}^\top \mathbf{P}_j \mathbf{x}_i) \cdot \sum_{j'=1}^{k} \xi^{-1}(v_{j'} - v_{j'}')\mathbf{a}^\top \mathbf{P}_{j'} \mathbf{x}_i,$$

$$I_4 = -\frac{1}{n} \sum_{i=1}^{n} \epsilon_i \sum_{j'=1}^{k} \xi^{-1}(v_{j'} - v_{j'}')\mathbf{a}^\top \mathbf{P}_{j'} \mathbf{x}_i.$$

By Lemma B.2 and Lemma B.6 we have

$$
\begin{aligned}
I_1 &\leq C_2 \xi^{-1} L\sqrt{k}\|\mathbf{v}^*\|_2\|\mathbf{v} - \mathbf{v}'\|_2, \\
I_2, I_3 &\leq C_2 D_0 \xi^{-1} L\sqrt{k}\|\mathbf{v} - \mathbf{v}'\|_2, \\
I_4 &\leq C_2 \nu \xi^{-1}\|\mathbf{v} - \mathbf{v}'\|_2
\end{aligned}
$$

for all $\mathbf{a} \in S^{r-1}$, $\mathbf{w} \in \mathcal{W}_0$ and $\mathbf{v}, \mathbf{v}' \in \mathcal{V}_0$, where $C_2$ is an absolute constant. Since $2\|\mathbf{v}^*\|_2 + D_0 \leq 3D_0$, we have

$$H_{wv} = \sup_{\substack{\mathbf{a} \in S^{r-1}, \mathbf{w} \in \mathcal{W}_0 \\ \mathbf{v}, \mathbf{v}' \in \mathcal{V}_0}} \frac{\mathbf{a}^\top [\mathbf{g}_w(\mathbf{w}, \mathbf{v}) - \mathbf{g}_w(\mathbf{w}, \mathbf{v}')]}{\|\mathbf{v} - \mathbf{v}'\|_2} \leq C_3 \xi^{-1}(\nu + D_0 L\sqrt{k}),$$

where $C_3$ is an absolute constant.

**Proof of** (5.3) By definition we have

$$
\begin{aligned}
H_{vw} &= \sup_{\substack{\mathbf{a}\in S^{k-1},\mathbf{v}\in\mathcal{V}_0 \\ \mathbf{w},\mathbf{w}'\in\mathcal{W}_0}} \frac{\mathbf{a}^\top[\mathbf{g}_v(\mathbf{w},\mathbf{v}) - \mathbf{g}_v(\mathbf{w}',\mathbf{v})]}{\|\mathbf{w}-\mathbf{w}'\|_2} \\
&= \sup_{\substack{\mathbf{a}\in S^{k-1},\mathbf{v}\in\mathcal{V}_0 \\ \mathbf{w},\mathbf{w}'\in\mathcal{W}_0}} -\frac{\mathbf{a}^\top\{[\mathbf{\Sigma}(\mathbf{w}) - \mathbf{\Sigma}(\mathbf{w}')]^\top\mathbf{y} + [\mathbf{\Sigma}^\top(\mathbf{w})\mathbf{\Sigma}(\mathbf{w}) - \mathbf{\Sigma}^\top(\mathbf{w}')\mathbf{\Sigma}(\mathbf{w}')]\mathbf{v}\}}{n\|\mathbf{w}-\mathbf{w}'\|_2} \\
&\le I_1' + I_2' + I_3',
\end{aligned}
$$

where

$$
I_1' = \sup_{\substack{\mathbf{a}\in S^{k-1}, \\ \mathbf{w},\mathbf{w}'\in\mathcal{W}_0}} -\frac{1}{n}\sum_{i=1}^{n}\sum_{j=1}^{k} \frac{a_j[\sigma(\mathbf{w}^\top\mathbf{P}_j\mathbf{x}_i) - \sigma(\mathbf{w}'^\top\mathbf{P}_j\mathbf{x}_i)]}{\|\mathbf{w}-\mathbf{w}'\|_2} \cdot \sum_{j'=1}^{k} v_{j'}^*\sigma(\mathbf{w}^{*\top}\mathbf{P}_{j'}\mathbf{x}_i),
$$

$$
I_2' = \sup_{\substack{\mathbf{a}\in S^{k-1},\mathbf{v}\in\mathcal{V}_0 \\ \mathbf{w},\mathbf{w}'\in\mathcal{W}_0}} -\frac{1}{n}\sum_{i=1}^{n}\sum_{j=1}^{k} a_j\sigma(\mathbf{w}^\top\mathbf{P}_j\mathbf{x}_i) \cdot \sum_{j'=1}^{k} \frac{v_{j'}[\sigma(\mathbf{w}^\top\mathbf{P}_{j'}\mathbf{x}_i) - \sigma(\mathbf{w}'^\top\mathbf{P}_{j'}\mathbf{x}_i)]}{\|\mathbf{w}-\mathbf{w}'\|_2},
$$

$$
I_3' = \sup_{\substack{\mathbf{a}\in S^{k-1},\mathbf{v}\in\mathcal{V}_0 \\ \mathbf{w},\mathbf{w}'\in\mathcal{W}_0}} -\frac{1}{n}\sum_{i=1}^{n}\sum_{j=1}^{k} \frac{a_j[\sigma(\mathbf{w}^\top\mathbf{P}_j\mathbf{x}_i) - \sigma(\mathbf{w}'^\top\mathbf{P}_j\mathbf{x}_i)]}{\|\mathbf{w}-\mathbf{w}'\|_2} \cdot \sum_{j'=1}^{k} v_{j'}\sigma(\mathbf{w}'^\top\mathbf{P}_{j'}\mathbf{x}_i),
$$

$$
I_4' = \sup_{\substack{\mathbf{a}\in S^{k-1}, \\ \mathbf{w},\mathbf{w}'\in\mathcal{W}_0}} -\frac{1}{n}\sum_{i=1}^{n}\epsilon_i \cdot \sum_{j=1}^{k} \frac{a_j[\sigma(\mathbf{w}^\top\mathbf{P}_j\mathbf{x}_i) - \sigma(\mathbf{w}'^\top\mathbf{P}_j\mathbf{x}_i)]}{\|\mathbf{w}-\mathbf{w}'\|_2}.
$$

Therefore by the Lipschitz continuity of $\sigma(\cdot)$, Lemma B.4 and Lemma B.7, we have

$$
I_1' \le \sup_{\substack{\mathbf{a}\in S^{k-1}, \\ \mathbf{w},\mathbf{w}'\in\mathcal{W}_0}} \frac{1}{n}\sum_{i=1}^{n}\sum_{j=1}^{k} \left| a_j\frac{(\mathbf{w}-\mathbf{w}')^\top}{\|\mathbf{w}-\mathbf{w}'\|_2}\mathbf{P}_j\mathbf{x}_i \right| \cdot \sum_{j'=1}^{k} |v_{j'}^*\sigma(\mathbf{w}^{*\top}\mathbf{P}_{j'}\mathbf{x}_i)| \le C_4 Lk\|\mathbf{v}^*\|_2,
$$

$$
I_2' \le \sup_{\substack{\mathbf{a}\in S^{k-1},\mathbf{v}\in\mathcal{V}_0 \\ \mathbf{w},\mathbf{w}'\in\mathcal{W}_0}} \frac{1}{n}\sum_{i=1}^{n}\sum_{j=1}^{k} |a_j\sigma(\mathbf{w}^\top\mathbf{P}_j\mathbf{x}_i)| \cdot \sum_{j'=1}^{k} \left| v_{j'}\frac{(\mathbf{w}-\mathbf{w}')^\top}{\|\mathbf{w}-\mathbf{w}'\|_2}\mathbf{P}_{j'}\mathbf{x}_i \right| \le C_4 LD_0 k,
$$

$$
I_3' \le \sup_{\substack{\mathbf{a}\in S^{k-1},\mathbf{v}\in\mathcal{V}_0 \\ \mathbf{w},\mathbf{w}'\in\mathcal{W}_0}} \frac{1}{n}\sum_{i=1}^{n}\sum_{j=1}^{k} \left| a_j\frac{(\mathbf{w}-\mathbf{w}')^\top}{\|\mathbf{w}-\mathbf{w}'\|_2}\mathbf{P}_j\mathbf{x}_i \right| \cdot \sum_{j'=1}^{k} |v_{j'}\sigma(\mathbf{w}'^\top\mathbf{P}_{j'}\mathbf{x}_i)| \le C_4 LD_0 k,
$$

$$
I_4' \le \sup_{\substack{\mathbf{a}\in S^{k-1}, \\ \mathbf{w},\mathbf{w}'\in\mathcal{W}_0}} \frac{1}{n}\sum_{i=1}^{n} |\epsilon_i| \cdot \sum_{j=1}^{k} \left| a_j\frac{(\mathbf{w}-\mathbf{w}')^\top}{\|\mathbf{w}-\mathbf{w}'\|_2}\mathbf{P}_j\mathbf{x}_i \right| \le C_4 \nu\sqrt{k},
$$

where $C_4$ is an absolute constant. Since $\|\mathbf{v}^*\|_2 \le D_0$, we have

$$
H_{vw} \le C_5(\nu + D_0 L\sqrt{k})\sqrt{k},
$$

where $C_5$ is an absolute constant.

**Proof of** (5.4) By Lemma B.5 we have

$$
\begin{aligned}
H_{vv} &= \sup_{\substack{\mathbf{a}\in S^{k-1},\mathbf{w}\in\mathcal{W}_0 \\ \mathbf{v},\mathbf{v}'\in\mathcal{V}_0}} \frac{\mathbf{a}^\top[\mathbf{g}_v(\mathbf{w},\mathbf{v})-\mathbf{g}_v(\mathbf{w},\mathbf{v}')]}{\|\mathbf{v}-\mathbf{v}'\|_2} \\
&= \sup_{\substack{\mathbf{a}\in S^{k-1},\mathbf{w}\in\mathcal{W}_0 \\ \mathbf{v},\mathbf{v}'\in\mathcal{V}_0}} -\frac{\mathbf{a}^\top\boldsymbol{\Sigma}^\top(\mathbf{w})\boldsymbol{\Sigma}(\mathbf{w})(\mathbf{v}-\mathbf{v}')}{n\|\mathbf{v}-\mathbf{v}'\|_2} \\
&= \sup_{\substack{\mathbf{a}\in S^{k-1},\mathbf{w}\in\mathcal{W}_0 \\ \mathbf{v},\mathbf{v}'\in\mathcal{V}_0}} -\frac{1}{n}\sum_{i=1}^{n}\sum_{j=1}^{k} a_j\sigma(\mathbf{w}^\top\mathbf{P}_j\mathbf{x}_i)\cdot\sum_{j'=1}^{k}\frac{v_{j'}-v'_{j'}}{\|\mathbf{v}-\mathbf{v}'\|_2}\sigma(\mathbf{w}^\top\mathbf{P}_{j'}\mathbf{x}_i) \\
&\le C_6 L^2 k,
\end{aligned}
$$

where $C_6$ is an absolute constant. This completes the proof of Lemma 5.2. $\qquad\square$

## B.3  Proof of Lemma 5.3

We first introduce the following lemma.

**Lemma B.8.** Let $z$ be a standard Gaussian random variable. Then under Assumption 3.1, $\sigma(z)$ is sub-Gaussian with

$$\|\sigma(z)\|_{\psi_2}\le CL \text{ and } \|\sigma(z)-\kappa\|_{\psi_2}\le C\Gamma,$$

where $C$ is an absolute constant, $L=1+|\sigma(0)|$ and $\Gamma=1+|\sigma(0)-\kappa|$.

*Proof of Lemma 5.3.* By assumption, with probability at least $1-\delta/3$, the bounds given in Lemma 5.2 all hold. let $\mathcal{N}_1=\mathcal{N}[\mathcal{W}_0,(kn)^{-1}]$, $\mathcal{N}_2=\mathcal{N}[\mathcal{V}_0,(kn)^{-1}]$ be $(kn)^{-1}$-nets covering $\mathcal{W}_0$ and $\mathcal{V}_0$ respectively. Then by the proof of Lemma 5.2 in [36], we have

$$|\mathcal{N}_1|\le(3kn)^r,\ |\mathcal{N}_2|\le(3kn)^k.$$

For any $\mathbf{w}\in\mathcal{W}_0$ and $\mathbf{v}\in\mathcal{V}_0$, there exists $\widehat{\mathbf{w}}\in\mathcal{N}_1$ and $\widehat{\mathbf{v}}\in\mathcal{N}_2$ such that

$$\|\mathbf{w}-\widehat{\mathbf{w}}\|_2\le(kn)^{-1},\ \|\mathbf{v}-\widehat{\mathbf{v}}\|_2\le(kn)^{-1}.$$

**Proof of** (5.5). By triangle inequality we have

$$\|\mathbf{g}_w(\mathbf{w},\mathbf{v})-\overline{\mathbf{g}}_w(\mathbf{w},\mathbf{v})\|_2\le A_1+A_2+A_3,$$

where

$$
\begin{aligned}
A_1 &= \|\mathbf{g}_w(\mathbf{w},\mathbf{v})-\mathbf{g}_w(\widehat{\mathbf{w}},\widehat{\mathbf{v}})\|_2, \\
A_2 &= \|\mathbf{g}_w(\widehat{\mathbf{w}},\widehat{\mathbf{v}})-\overline{\mathbf{g}}_w(\widehat{\mathbf{w}},\widehat{\mathbf{v}})\|_2, \\
A_3 &= \|\overline{\mathbf{g}}_w(\widehat{\mathbf{w}},\widehat{\mathbf{v}})-\overline{\mathbf{g}}_w(\mathbf{w},\mathbf{v})\|_2.
\end{aligned}
$$

For $A_1$, we have

$$
\begin{aligned}
A_1 &\le \|\mathbf{g}_w(\mathbf{w},\mathbf{v})-\mathbf{g}_w(\widehat{\mathbf{w}},\mathbf{v})\|_2+\|\mathbf{g}_w(\widehat{\mathbf{w}},\mathbf{v})-\mathbf{g}_w(\widehat{\mathbf{w}},\widehat{\mathbf{v}})\|_2 \\
&\le C_1\xi^{-1}D_0^2\sqrt{k}\|\mathbf{w}-\widehat{\mathbf{w}}\|_2+C_2\xi^{-1}(\nu+D_0L\sqrt{k})\|\mathbf{v}-\widehat{\mathbf{v}}\|_2 \\
&\le C_3\xi^{-1}(\nu+D_0^2+D_0L)\frac{1}{n\sqrt{k}},
\end{aligned}
\tag{B.1}
$$

where $C_1$, $C_2$ and $C_3$ are absolute constants. For $A_2$, by direct calculation we have

$$\overline{\mathbf{g}}_w(\widehat{\mathbf{w}},\widehat{\mathbf{v}})=\mathbb{E}[\mathbf{g}_w(\widehat{\mathbf{w}},\widehat{\mathbf{v}})].$$

Let $\mathcal{N}_3=\mathcal{N}(S^{r-1},1/2)$ be a $1/2$-net covering $S^{r-1}$. Then by Lemma 5.2 in [36] we have $|\mathcal{N}_3|\le 5^r$. By definition, for any $\mathbf{a}\in\mathcal{N}_3$ we have

$$\mathbf{a}^\top\mathbf{g}_w(\widehat{\mathbf{w}},\widehat{\mathbf{v}})=-\frac{1}{n\xi}\sum_{i=1}^{n}[U_i^*-U_i+\epsilon_i+\kappa\mathbf{1}^\top(\mathbf{v}^*-\widehat{\mathbf{v}})]V_i,$$

where

$$U_i^* = \sum_{j=1}^{k} v_j^*[\sigma(\mathbf{w}^{*\top}\mathbf{P}_j\mathbf{x}_i) - \kappa], \; U_i = \sum_{j=1}^{k} \widehat{v}_j[\sigma(\widehat{\mathbf{w}}^\top\mathbf{P}_j\mathbf{x}_i) - \kappa], \; V_i = \sum_{j=1}^{k} \widehat{v}_j\mathbf{a}^\top\mathbf{P}_j\mathbf{x}_i.$$

By Lemma B.8, $\sigma(\mathbf{w}^{*\top}\mathbf{P}_j\mathbf{x}_i) - \kappa$ and $\sigma(\widehat{\mathbf{w}}^\top\mathbf{P}_j\mathbf{x}_i) - \kappa$ are centered sub-Gaussian random variables with $\|\sigma(\mathbf{w}^{*\top}\mathbf{P}_j\mathbf{x}_i) - \kappa\|_{\psi_2}, \|\sigma(\widehat{\mathbf{w}}^\top\mathbf{P}_j\mathbf{x}_i) - \kappa\|_{\psi_2} \le C_4\Gamma$ for some absolute constant $C_4$. Therefore by Lemma 5.9 in [36], we have $\|U_i^*\|_{\psi_2} \le C_5\|\mathbf{v}^*\|_2\Gamma$ and $\|U_i\|_{\psi_2} \le C_5 D_0\Gamma$, where $C_5$ is an absolute constant. Similarly, we have $\|V_i\|_{\psi_2} \le C_6 D_0$ for some absolute constant $C_6$. Therefore, by Lemma E.1 we have

$$\begin{aligned}
\|[U_i^* - U_i + \epsilon_i + \kappa\mathbf{1}^\top(\mathbf{v}^* - \widehat{\mathbf{v}})]V_i\|_{\psi_1} &\le C_7 D_0[(D_0 + \|\mathbf{v}^*\|_2)\Gamma + M + \nu] \\
&\le C_8 D_0(D_0\Gamma + M + \nu),
\end{aligned}$$

where $C_7$ and $C_8$ are absolute constants. By Proposition 5.16 in [36], with probability at least $1 - \delta/3$, we have

$$|\mathbf{a}^\top[\mathbf{g}_w(\widehat{\mathbf{w}}, \widehat{\mathbf{v}}) - \overline{\mathbf{g}}_w(\widehat{\mathbf{w}}, \widehat{\mathbf{v}})]| \le C_9\xi^{-1}D_0(D_0\Gamma + M + \nu)\sqrt{\frac{(r+k)\log(90nk/\delta)}{n}}$$

for all $\widehat{\mathbf{w}} \in \mathcal{N}_1, \widehat{\mathbf{v}} \in \mathcal{N}_2$ and $\mathbf{a} \in \mathcal{N}_3$, where $C_9$ is an absolute constant. Therefore by Lemma 5.3 in [36], we have

$$A_2 \le C_{10}\xi^{-1}D_0(D_0\Gamma + M + \nu)\sqrt{\frac{(r+k)\log(90nk/\delta)}{n}} \tag{B.2}$$

for all $\widehat{\mathbf{w}} \in \mathcal{N}_1, \widehat{\mathbf{v}} \in \mathcal{N}_2$, where $C_{10}$ is an absolute constant. For $A_3$, by triangle inequality we have

$$\begin{aligned}
A_3 &\le \|\overline{\mathbf{g}}_w(\widehat{\mathbf{w}}, \widehat{\mathbf{v}}) - \overline{\mathbf{g}}_w(\mathbf{w}, \widehat{\mathbf{v}})\|_2 + \|\overline{\mathbf{g}}_w(\mathbf{w}, \widehat{\mathbf{v}}) - \overline{\mathbf{g}}_w(\mathbf{w}, \mathbf{v})\|_2 \\
&\le \|\widehat{\mathbf{v}}\|_2^2\|\mathbf{w} - \widehat{\mathbf{w}}\|_2 + \left[\left|\|\mathbf{v}\|_2^2 - \|\widehat{\mathbf{v}}\|_2^2\right| + \left|\mathbf{v}^{*\top}(\mathbf{v} - \widehat{\mathbf{v}})\right|\right] \\
&\le D_0^2\|\mathbf{w} - \widehat{\mathbf{w}}\|_2 + 3D_0\|\mathbf{v} - \widehat{\mathbf{v}}\|_2 \\
&\le 4D_0(D_0 + 1)(nk)^{-1}. \tag{B.3}
\end{aligned}$$

By (B.1), (B.2), (B.3), and the assumptions on sample size $n$, we have

$$\|\mathbf{g}_w(\mathbf{w}, \mathbf{v}) - \overline{\mathbf{g}}_w(\mathbf{w}, \mathbf{v})\|_2 \le C_{11}\xi^{-1}D_0(D_0\Gamma + M + \nu)\sqrt{\frac{(r+k)\log(90nk/\delta)}{n}},$$

where $C_{11}$ is an absolute constant.

**Proof of** (5.6). By triangle inequality we have

$$\|\mathbf{g}_v(\mathbf{w}, \mathbf{v}) - \overline{\mathbf{g}}_v(\mathbf{w}, \mathbf{v})\|_2 \le B_1 + B_2 + B_3,$$

where

$$\begin{aligned}
B_1 &= \|\mathbf{g}_v(\mathbf{w}, \mathbf{v}) - \mathbf{g}_v(\widehat{\mathbf{w}}, \widehat{\mathbf{v}})\|_2, \\
B_2 &= \|\mathbf{g}_v(\widehat{\mathbf{w}}, \widehat{\mathbf{v}}) - \overline{\mathbf{g}}_v(\widehat{\mathbf{w}}, \widehat{\mathbf{v}})\|_2, \\
B_3 &= \|\overline{\mathbf{g}}_v(\widehat{\mathbf{w}}, \widehat{\mathbf{v}}) - \overline{\mathbf{g}}_v(\mathbf{w}, \mathbf{v})\|_2.
\end{aligned}$$

For $B_1$, by Lemma 5.2 we have

$$\begin{aligned}
B_1 &\le \|\mathbf{g}_v(\mathbf{w}, \mathbf{v}) - \mathbf{g}_v(\widehat{\mathbf{w}}, \mathbf{v})\|_2 + \|\mathbf{g}_v(\widehat{\mathbf{w}}, \mathbf{v}) - \mathbf{g}_v(\widehat{\mathbf{w}}, \widehat{\mathbf{v}})\|_2 \\
&\le C_{12}(\nu + D_0 L\sqrt{k})\sqrt{k}\|\mathbf{w} - \widehat{\mathbf{w}}\|_2 + C_{13}L^2 k\|\mathbf{v} - \widehat{\mathbf{v}}\|_2 \\
&\le C_{14}(\nu\sqrt{k} + D_0 Lk + L^2 k)/(nk), \tag{B.4}
\end{aligned}$$

where $C_{12}, C_{13}$ and $C_{14}$ are absolute constants. For $B_2$, by direct calculation we have

$$\overline{\mathbf{g}}_v(\widehat{\mathbf{w}}, \widehat{\mathbf{v}}) = \mathbb{E}[\mathbf{g}_v(\widehat{\mathbf{w}}, \widehat{\mathbf{v}})].$$

Let $\mathcal{N}_4 = \mathcal{N}(S^{k-1}, 1/2)$ be a $1/2$-net covering $S^{k-1}$. Then by Lemma 5.2 in [36] we have $|\mathcal{N}_4| \le 5^k$. By definition, for any $\mathbf{b} \in \mathcal{N}_4$ we have

$$\mathbf{b}^\top\mathbf{g}_v(\widehat{\mathbf{w}}, \widehat{\mathbf{v}}) = -\frac{1}{n}\sum_{i=1}^{n}[U_i^* - U_i + \epsilon_i + \kappa\mathbf{1}^\top(\mathbf{v}^* - \widehat{\mathbf{v}})](U_i' + \kappa\mathbf{1}^\top\mathbf{b}),$$

where

$$U_i^* = \sum_{j=1}^k v_j^*[\sigma(\mathbf{w}^{*\top}\mathbf{P}_j\mathbf{x}_i) - \kappa], \; U_i = \sum_{j=1}^k \widehat{v}_j^t[\sigma(\widehat{\mathbf{w}}^\top\mathbf{P}_j\mathbf{x}_i) - \kappa], \; U_i' = \sum_{j=1}^k b_j[\sigma(\widehat{\mathbf{w}}^\top\mathbf{P}_j\mathbf{x}_i) - \kappa].$$

Similar to the proof of (5.5), we have $\|U_i^*\|_{\psi_2} \leq C_{15}\|\mathbf{v}^*\|_2\Gamma$, $\|U_i\|_{\psi_2} \leq C_{15}D_0\Gamma$, and $\|U_i'\|_{\psi_2} \leq C_{15}\Gamma$, where $C_{15}$ is an absolute constant. Therefore by Lemma E.1, we have

$$\|[U_i^* - U_i + \epsilon_i + \kappa\mathbf{1}^\top(\mathbf{v}^* - \widehat{\mathbf{v}})](U_i' + \kappa\mathbf{1}^\top\mathbf{b})\|_{\psi_2} \leq C_{16}(\Gamma + \kappa\sqrt{k})[D_0\Gamma + M + \nu],$$

where $C_{16}$ is an absolute constant. By Proposition 5.16 in [36], with probability at least $1 - \delta/4$ we have

$$|\mathbf{b}^\top[\mathbf{g}_v(\widehat{\mathbf{w}}, \widehat{\mathbf{v}}) - \overline{\mathbf{g}}_v(\widehat{\mathbf{w}}, \widehat{\mathbf{v}})]| \leq C_{17}(\Gamma + \kappa\sqrt{k})[D_0\Gamma + M + \nu]\sqrt{\frac{(r+k)\log(90nk/\delta)}{n}}$$

for all $\widehat{\mathbf{w}} \in \mathcal{N}_1, \widehat{\mathbf{v}} \in \mathcal{N}_2$ and $\mathbf{b} \in \mathcal{N}_4$, where $C_{17}$ is an absolute constant. By Lemma 5.3 in [36], we have

$$\|\mathbf{g}_v(\widehat{\mathbf{w}}, \widehat{\mathbf{v}}) - \overline{\mathbf{g}}_v(\widehat{\mathbf{w}}, \widehat{\mathbf{v}})\|_2 \leq C_{18}(\Gamma + \kappa\sqrt{k})[D_0\Gamma + M + \nu]\sqrt{\frac{(r+k)\log(90nk/\delta)}{n}} \tag{B.5}$$

for all $\widehat{\mathbf{w}} \in \mathcal{N}_1, \widehat{\mathbf{v}} \in \mathcal{N}_2$, where $C_{18}$ is an absolute constant. For $B_3$, by definition and Lemma B.9 we have

$$\begin{aligned} B_3 &\leq (\Delta + \kappa k)\|\mathbf{v} - \widehat{\mathbf{v}}\|_2 + \|\mathbf{v}^*\|_2|\phi(\mathbf{w}, \mathbf{w}^*) - \phi(\widehat{\mathbf{w}}, \mathbf{w}^*)| \\ &\leq (\Delta + \kappa k)\|\mathbf{v} - \widehat{\mathbf{v}}\|_2 + L\|\mathbf{v}^*\|_2\|\mathbf{w} - \widehat{\mathbf{w}}\|_2 \\ &\leq (\Delta + \kappa k + D_0 L)(nk)^{-1}. \end{aligned} \tag{B.6}$$

By (B.4), (B.5), (B.6) and the assumptions on the sample size $n$, we have

$$\|\mathbf{g}_v(\mathbf{w}, \mathbf{v}) - \overline{\mathbf{g}}_v(\mathbf{w}, \mathbf{v})\|_2 \leq C_{19}(\Gamma + \kappa\sqrt{k})[D_0\Gamma + M + \nu]\sqrt{\frac{(r+k)\log(90nk/\delta)}{n}},$$

where $C_{19}$ is an absolute constant. This completes the proof of (5.6). $\square$

## B.4 Proof of Lemma 5.4

We remind the readers that $L = 1 + |\sigma(0)|$ and $\Gamma = 1 + |\sigma(0) - \kappa|$ are constants that only depends on the activation function. We introduce the following notations. Let

$$\begin{aligned} \overline{\mathbf{g}}_w^t &= \overline{\mathbf{g}}_w(\mathbf{w}^t, \mathbf{v}^t), \quad \overline{\mathbf{g}}_v^t = \overline{\mathbf{g}}_v(\mathbf{w}^t, \mathbf{v}^t), \\ \overline{\mathbf{u}}^{t+1} &= \mathbf{w}^t - \alpha\overline{\mathbf{g}}_w^t, \quad \overline{\mathbf{v}}^{t+1} = \mathbf{v}^t - \alpha\overline{\mathbf{g}}_v^t. \end{aligned}$$

Then it is easy to see that for any fixed $\mathbf{w}^t, \mathbf{v}^t$, the vectors $\overline{\mathbf{g}}_w^t, \overline{\mathbf{g}}_v^t, \overline{\mathbf{u}}^{t+1}, \overline{\mathbf{v}}^{t+1}$ defined above are expectations of $\mathbf{g}_w^t, \mathbf{g}_v^t, \mathbf{u}^{t+1}, \mathbf{v}^{t+1}$ respectively. We will also use the result of the following lemma.

**Lemma B.9.** Under Assumption 3.1, for any fixed unit vector $\mathbf{w}$, $\phi(\mathbf{w}, \cdot)$ is Lipschitz continuous with Lipschitz constant $L = 1 + |\sigma(0)|$.

*Proof of Lemma 5.4.* We now list the proof of (5.7)-(5.10) as follows.

**Proof of (5.7).** By the definition of $\overline{\mathbf{u}}^{t+1}$ and $\overline{\mathbf{g}}^t$ we have

$$\overline{\mathbf{u}}^{t+1} = \mathbf{w}^t - \alpha\overline{\mathbf{g}}_w^t = (1 - \alpha\|\mathbf{v}^t\|_2^2)\mathbf{w}^t + \alpha(\mathbf{v}^{*\top}\mathbf{v}^t)\mathbf{w}^*.$$

The equation above implies that when $1 - \alpha\|\mathbf{v}^t\|_2^2 \geq 0$, $\overline{\mathbf{u}}^{t+1}$ is in the cone spanned by $\mathbf{w}^t$ and $\mathbf{w}^*$. To simplify notation, we define $\widehat{\mathbf{u}} = \overline{\mathbf{u}}^{t+1}/\|\overline{\mathbf{u}}^{t+1}\|_2$. By $1 - \alpha\|\mathbf{v}^t\|_2^2 < 1$ and $\alpha(\mathbf{v}^{*\top}\mathbf{v}^t) \geq \alpha\rho$, we have

$$\mathbf{w}^{*\top}\widehat{\mathbf{u}} \geq \mathbf{w}^{*\top}\frac{\alpha\rho\mathbf{w}^* + \mathbf{w}^t}{\|\alpha\rho\mathbf{w}^* + \mathbf{w}^t\|_2} = \frac{\alpha\rho + \mathbf{w}^{*\top}\mathbf{w}^t}{\|\alpha\rho\mathbf{w}^* + \mathbf{w}^t\|_2} \geq \frac{\alpha\rho + \mathbf{w}^{*\top}\mathbf{w}^t}{1 + \alpha\rho}. \tag{B.7}$$

The first inequality in (B.7) is further explained in Figure 2. Moreover, by Lemma 5.3 we have

Figure 2: Explanation of (B.7). The two arrows denotes $\mathbf{w}^*$ and $\mathbf{w}^t$. The gray area shows all possible positions of $a\mathbf{w}^* + b\mathbf{w}^t$ with $a \geq \alpha\rho$ and $0 \leq b \leq 1$. We use blue and green dots to represent the worst case of $\overline{\mathbf{u}}^{t+1}$ and $\widehat{\mathbf{u}}$ respectively. The first inequality in (B.7) is then easily obtained by replacing $\widehat{\mathbf{u}}$ with its worst case value, which is $\frac{\alpha\rho\mathbf{w}^* + \mathbf{w}^t}{\|\alpha\rho\mathbf{w}^* + \mathbf{w}^t\|_2}$.

$$\|\mathbf{u}^{t+1} - \overline{\mathbf{u}}^{t+1}\|_2 = \alpha\|\mathbf{g}_w^t - \overline{\mathbf{g}}_w^t\|_2 \leq \alpha\eta_w.$$

By Lemma E.2, we have

$$\|\mathbf{w}^{t+1} - \widehat{\mathbf{u}}\|_2 \leq \frac{2\alpha\eta_w}{\|\overline{\mathbf{u}}^{t+1}\|_2}.$$

By $\mathbf{w}^{*\top}\mathbf{w}^t > 0$, we have

$$\|\overline{\mathbf{u}}^{t+1}\|_2 = \left\|(1 - \alpha\|\mathbf{v}^t\|_2^2)\mathbf{w}^t + \alpha\mathbf{v}^{t\top}\mathbf{v}^*\mathbf{w}^*\right\|_2 \geq 1 - \alpha\|\mathbf{v}^t\|_2^2 \geq 1 - \alpha D_0^2 \geq \frac{1}{2}.$$

Therefore,

$$\|\mathbf{w}^{t+1} - \widehat{\mathbf{u}}\|_2 \leq 4\alpha\eta_w. \tag{B.8}$$

Since $\mathbf{w}^*$, $\mathbf{w}^t$ and $\widehat{u}$ are all unit vectors, by (B.7) we have

$$1 - \frac{1}{2}\|\mathbf{w}^* - \widehat{\mathbf{u}}\|_2^2 \geq \frac{\alpha\rho}{1 + \alpha\rho} + \frac{1}{1 + \alpha\rho}\left(1 - \frac{1}{2}\|\mathbf{w}^{*\top} - \mathbf{w}^t\|_2^2\right).$$

Rearranging terms gives

$$\|\widehat{\mathbf{u}} - \mathbf{w}^*\|_2 \leq \frac{1}{\sqrt{1 + \alpha\rho}}\|\mathbf{w}^t - \mathbf{w}^*\|_2.$$

By (B.8) we have

$$\|\mathbf{w}^{t+1} - \mathbf{w}^*\|_2 \leq \frac{1}{\sqrt{1 + \alpha\rho}}\|\mathbf{w}^t - \mathbf{w}^*\|_2 + 4\alpha\eta_w \leq \frac{1}{\sqrt{1 + \alpha\rho}}\|\mathbf{w}^t - \mathbf{w}^*\|_2 + \frac{8\alpha\sqrt{1 + \alpha\rho}}{1 + \sqrt{1 + \alpha\rho}}\eta_w.$$

Rearranging terms again, we obtain

$$\|\mathbf{w}^{t+1} - \mathbf{w}^*\|_2 - 8\rho^{-1}(1 + \alpha\rho)\eta_w \leq \frac{1}{\sqrt{1 + \alpha\rho}}[\|\mathbf{w}^t - \mathbf{w}^*\|_2 - 8\rho^{-1}(1 + \alpha\rho)\eta_w].$$

This completes the proof of (5.7).

**Proof of** (5.8). By the definition of $\overline{\mathbf{g}}_v^t$ we have

$$\begin{aligned}
\mathbf{1}^\top\overline{\mathbf{g}}_v^t &= \mathbf{1}^\top\{(\Delta\mathbf{I} + \kappa^2\mathbf{1}\mathbf{1}^\top)\mathbf{v}^t - [\phi(\mathbf{w}^t, \mathbf{w}^*)\mathbf{I} + \kappa^2\mathbf{1}\mathbf{1}^\top]\mathbf{v}^*\} \\
&= (\Delta + \kappa^2 k)\mathbf{1}^\top\mathbf{v}^t - [\phi(\mathbf{w}^t, \mathbf{w}^*) + \kappa^2 k]\mathbf{1}^\top\mathbf{v}^* \\
&= (\Delta + \kappa^2 k)\mathbf{1}^\top(\mathbf{v}^t - \mathbf{v}^*) + [\Delta - \phi(\mathbf{w}^t, \mathbf{w}^*)]\mathbf{1}^\top\mathbf{v}^*.
\end{aligned}$$

Therefore,

$$\mathbf{1}^\top(\overline{\mathbf{v}}^{t+1} - \mathbf{v}^*) = [1 - \alpha(\Delta + \kappa^2 k)]\mathbf{1}^\top(\mathbf{v}^t - \mathbf{v}^*) - \alpha[\Delta - \phi(\mathbf{w}^t, \mathbf{w}^*)]\mathbf{1}^\top\mathbf{v}^*.$$

By Lemma 5.3, $|\mathbf{1}^\top(\mathbf{v}^{t+1} - \mathbf{v}^t)| \le \alpha|\mathbf{1}^\top(\mathbf{g}^t - \overline{\mathbf{g}}^t)| \le \alpha\sqrt{k}\|\mathbf{g}^t - \overline{\mathbf{g}}^t\|_2 \le \alpha\sqrt{k}\eta_v$. Therefore by triangle inequality we have

$$|\mathbf{1}^\top(\mathbf{v}^{t+1} - \mathbf{v}^*)| \le [1 - \alpha(\Delta + \kappa^2 k)]|\mathbf{1}^\top(\mathbf{v}^t - \mathbf{v}^*)| + \alpha|\Delta - \phi(\mathbf{w}^t, \mathbf{w}^*)| \cdot |\mathbf{1}^\top\mathbf{v}^*| + \alpha\sqrt{k}\eta_v.$$

By lemma B.9, we have

$$|\Delta - \phi(\mathbf{w}^t, \mathbf{w}^*)| = |\phi(\mathbf{w}^*, \mathbf{w}^*) - \phi(\mathbf{w}^t, \mathbf{w}^*)| \le L\|\mathbf{w}^t - \mathbf{w}^*\|_2.$$

Therefore,

$$|\mathbf{1}^\top(\mathbf{v}^{t+1} - \mathbf{v}^*)| \le [1 - \alpha(\Delta + \kappa^2 k)]|\mathbf{1}^\top(\mathbf{v}^t - \mathbf{v}^*)| + \alpha L\|\mathbf{w}^t - \mathbf{w}^*\|_2|\mathbf{1}^\top\mathbf{v}^*| + \alpha\sqrt{k}\eta_v. \quad \text{(B.9)}$$

This completes the proof of (5.8).

**Proof of** (5.9). By the definition of $\overline{\mathbf{g}}_v^t$, we have

$$\begin{aligned}
\overline{\mathbf{g}}_v^t &= (\Delta\mathbf{I} + \kappa^2\mathbf{1}\mathbf{1}^\top)\mathbf{v}^t - [\phi(\mathbf{w}^t, \mathbf{w}^*)\mathbf{I} + \kappa^2\mathbf{1}\mathbf{1}^\top]\mathbf{v}^* \\
&= \Delta\mathbf{v}^t - \phi(\mathbf{w}^t, \mathbf{w}^*)\mathbf{v}^* + \kappa^2\mathbf{1}\mathbf{1}^\top(\mathbf{v}^t - \mathbf{v}^*) \\
&= \Delta(\mathbf{v}^t - \mathbf{v}^*) + [\Delta - \phi(\mathbf{w}^t, \mathbf{w}^*)]\mathbf{v}^* + \kappa^2\mathbf{1}\mathbf{1}^\top(\mathbf{v}^t - \mathbf{v}^*). \quad \text{(B.10)}
\end{aligned}$$

Therefore

$$\begin{aligned}
(\mathbf{v}^t - \mathbf{v}^*)^\top\overline{\mathbf{g}}_v^t &\ge \Delta\|\mathbf{v}^t - \mathbf{v}^*\|_2^2 + [\Delta - \phi(\mathbf{w}^t, \mathbf{w}^*)](\mathbf{v}^t - \mathbf{v}^*)^\top\mathbf{v}^* \\
&\ge \Delta\|\mathbf{v}^t - \mathbf{v}^*\|_2^2 - |\Delta - \phi(\mathbf{w}^t, \mathbf{w}^*)| \cdot \|\mathbf{v}^t - \mathbf{v}^*\|_2\|\mathbf{v}^*\|_2.
\end{aligned}$$

By Lemma B.9, we have

$$|\Delta - \phi(\mathbf{w}^t, \mathbf{w}^*)| = |\phi(\mathbf{w}^*, \mathbf{w}^*) - \phi(\mathbf{w}^t, \mathbf{w}^*)| \le L\|\mathbf{w}^* - \mathbf{w}^t\|_2.$$

Therefore,

$$\begin{aligned}
(\mathbf{v}^t - \mathbf{v}^*)^\top\overline{\mathbf{g}}_v^t &\ge \Delta\|\mathbf{v}^t - \mathbf{v}^*\|_2^2 - L\|\mathbf{w}^* - \mathbf{w}^t\|_2\|\mathbf{v}^t - \mathbf{v}^*\|_2\|\mathbf{v}^*\|_2 \\
&\ge \Delta\|\mathbf{v}^t - \mathbf{v}^*\|_2^2 - \frac{1}{2}\left(\frac{L^2}{\Delta}\|\mathbf{v}^*\|_2^2\|\mathbf{w}^* - \mathbf{w}^t\|_2^2 + \Delta\|\mathbf{v}^t - \mathbf{v}^*\|_2^2\right) \\
&\ge \frac{\Delta}{2}\|\mathbf{v}^t - \mathbf{v}^*\|_2^2 - \frac{L^2}{2\Delta}\|\mathbf{v}^*\|_2^2\|\mathbf{w}^* - \mathbf{w}^t\|_2^2.
\end{aligned}$$

By Lemma 5.3, we have

$$\begin{aligned}
(\mathbf{v}^t - \mathbf{v}^*)^\top\mathbf{g}_v^t &\ge \frac{\Delta}{2}\|\mathbf{v}^t - \mathbf{v}^*\|_2^2 - \frac{L^2}{2\Delta}\|\mathbf{v}^*\|_2^2\|\mathbf{w}^* - \mathbf{w}^t\|_2^2 - \|\mathbf{v}^t - \mathbf{v}^*\|_2\eta_v \\
&\ge \frac{\Delta}{4}\|\mathbf{v}^t - \mathbf{v}^*\|_2^2 - \frac{L^2}{2\Delta}\|\mathbf{v}^*\|_2^2\|\mathbf{w}^* - \mathbf{w}^t\|_2^2 - \frac{1}{\Delta}\eta_v^2
\end{aligned}$$

Moreover, by (B.10) we have

$$\|\overline{\mathbf{g}}_v^t\|_2 \le \Delta\|\mathbf{v}^t - \mathbf{v}^*\|_2 + L\|\mathbf{v}^*\|_2\|\mathbf{w}^t - \mathbf{w}^*\|_2 + \kappa^2\sqrt{k}|\mathbf{1}^\top(\mathbf{v}^t - \mathbf{v}^*)|.$$

By Lemma 5.3, we have

$$\|\mathbf{g}_v^t\|_2 \le \Delta\|\mathbf{v}^t - \mathbf{v}^*\|_2 + L\|\mathbf{v}^*\|_2\|\mathbf{w}^t - \mathbf{w}^*\|_2 + \kappa^2\sqrt{k}|\mathbf{1}^\top(\mathbf{v}^t - \mathbf{v}^*)| + \eta_v,$$

Therefore,

$$\begin{aligned}
\|\mathbf{v}^{t+1} - \mathbf{v}^*\|_2^2 &= \|\mathbf{v}^t - \alpha\mathbf{g}_v^t - \mathbf{v}^*\|_2^2 \\
&= \|\mathbf{v}^t - \mathbf{v}^*\|_2^2 - 2\alpha(\mathbf{v}^t - \mathbf{v}^*)^\top\mathbf{g}_v^t + \alpha^2\|\mathbf{g}_v^t\|_2^2.
\end{aligned}$$

Plugging in the previous inequalities gives

$$\begin{aligned}
\|\mathbf{v}^{t+1} - \mathbf{v}^*\|_2^2 \le &(1 - \alpha\Delta + 4\alpha^2\Delta^2)\|\mathbf{v}^t - \mathbf{v}^*\|_2^2 + \left(\frac{\alpha L^2}{\Delta} + 4\alpha^2 L^2\right)\|\mathbf{v}^*\|_2^2\|\mathbf{w}^* - \mathbf{w}^t\|_2^2 \\
&+ 4\alpha^2\kappa^4 k[\mathbf{1}^\top(\mathbf{v}^t - \mathbf{v}^*)]^2 + \left(\frac{2\alpha}{\Delta} + 4\alpha^2\right)\eta_v^2.
\end{aligned}$$

This completes the proof of (5.9).

**Proof of** (5.10). We first check $\mathbf{w}^{t+1} \in \mathcal{W}$. By (B.7) and (B.8) we have

$$\mathbf{w}^{*\top}\mathbf{w}^{t+1} \geq \frac{\alpha\rho + \mathbf{w}^{*\top}\mathbf{w}^t}{1+\alpha\rho} - 4\alpha\eta_w.$$

Since $\mathbf{w}^t \in \mathcal{W}$, we have $\mathbf{w}^{*\top}\mathbf{w}^0 \leq 2\mathbf{w}^{*\top}\mathbf{w}^t$. Moreover, by $\mathbf{w}^{*\top}\mathbf{w}^0 \leq 1$, we have

$$\mathbf{w}^{*\top}\mathbf{w}^{t+1} \geq \frac{\alpha\rho}{1+\alpha\rho}\mathbf{w}^{*\top}\mathbf{w}^0 + \frac{1}{2}\frac{1}{1+\alpha\rho}\mathbf{w}^{*\top}\mathbf{w}^0 - 4\alpha\eta_w \geq \frac{1}{2}\mathbf{w}^{*\top}\mathbf{w}^0.$$

where the last inequality follows by the assumption that

$$4\alpha\eta_w \leq \frac{1}{2}\frac{\alpha\rho}{1+\alpha\rho}\mathbf{w}^{*\top}\mathbf{w}^0.$$

Since $\|\mathbf{w}^{t+1}\|_2 = 1$, we have $\mathbf{w}^{t+1} \in \mathcal{W}$. For $\mathbf{v}^{t+1}$, since $\mathbf{v}^t \in \mathcal{V}$, by (5.9) and the definition of $M$ and $D$ we have

$$\begin{aligned}
\|\mathbf{v}^{t+1} - \mathbf{v}^*\|_2^2 &\leq (1 - \alpha\Delta + 4\alpha^2\Delta^2)\|\mathbf{v}^t - \mathbf{v}^*\|_2^2 + \alpha(\Delta^{-1} + 4\alpha)L^2\|\mathbf{v}^*\|_2^2 \cdot 4 \\
&\quad + 4\alpha^2\kappa^2 M^2 k + 2\alpha(\Delta^{-1} + 2\alpha) \\
&\leq (1 - \alpha\Delta + 4\alpha^2\Delta^2)D^2 + \alpha\Delta(1 - 4\alpha\Delta)D^2 \\
&= D^2.
\end{aligned}$$

Therefore we have $\|\mathbf{v}^{t+1} - \mathbf{v}^*\|_2 \leq D$. Since $\mathbf{v}^t \in \mathcal{V}$, by (B.9) and the definition of $M$ we have

$$\begin{aligned}
|\kappa\mathbf{1}^\top(\mathbf{v}^{t+1} - \mathbf{v}^*)| &\leq [1 - \alpha(\Delta + \kappa^2 k)]|\kappa\mathbf{1}^\top(\mathbf{v}^t - \mathbf{v}^*)| + \alpha|\kappa|L\|\mathbf{w}^t - \mathbf{w}^*\|_2|\mathbf{1}^\top\mathbf{v}^*| + \alpha|\kappa|\sqrt{k}\eta_v \\
&\leq [1 - \alpha(\Delta + \kappa^2 k)]M + 2\alpha|\kappa|L|\mathbf{1}^\top\mathbf{v}^*| + \alpha|\kappa|\sqrt{k} \\
&\leq [1 - \alpha(\Delta + \kappa^2 k)]M + \alpha(\Delta + \kappa^2 k)M \\
&= M.
\end{aligned}$$

We now check $\mathbf{v}^{*\top}\mathbf{v}^{t+1} \geq \rho$. By assumption we have $\kappa^2(\mathbf{1}^\top\mathbf{v}^*)\mathbf{1}^\top(\mathbf{v}^t - \mathbf{v}^*) \leq \rho$. Therefore by definition we have

$$\begin{aligned}
\mathbf{v}^{*\top}\overline{\mathbf{g}}_v^t &= \Delta\mathbf{v}^{*\top}\mathbf{v}^t + \kappa^2(\mathbf{1}^\top\mathbf{v}^*)(\mathbf{1}^\top\mathbf{v}^t) - \psi(\mathbf{w}^{*\top}\mathbf{w}^t)\|\mathbf{v}^*\|_2^2 - \kappa^2(\mathbf{1}^\top\mathbf{v}^*)^2 \\
&\leq \Delta\mathbf{v}^{*\top}\mathbf{v}^t - \psi(\mathbf{w}^{*\top}\mathbf{w}^t)\|\mathbf{v}^*\|_2^2 + \rho.
\end{aligned}$$

By Lemma 4.1, $\psi(\tau)$ is an increasing function. Therefore,

$$\begin{aligned}
\mathbf{v}^{*\top}\overline{\mathbf{v}}^{t+1} &= \mathbf{v}^{*\top}(\mathbf{v}^t - \alpha\overline{\mathbf{g}}_v^t) \\
&\geq \mathbf{v}^{*\top}\mathbf{v}^t - \alpha[\Delta\mathbf{v}^{*\top}\mathbf{v}^t - \psi(\mathbf{w}^{*\top}\mathbf{w}^t)\|\mathbf{v}^*\|_2^2 + \rho] \\
&\geq (1 - \alpha\Delta)\mathbf{v}^{*\top}\mathbf{v}^t + \alpha\psi(\mathbf{w}^{*\top}\mathbf{w}^0/2)\|\mathbf{v}^*\|_2^2 - \alpha\rho \\
&\geq (1 - \alpha\Delta)\rho + \alpha\psi(\mathbf{w}^{*\top}\mathbf{w}^0/2)\|\mathbf{v}^*\|_2^2 - \alpha\rho.
\end{aligned}$$

By the definition of $\rho$, we have

$$\psi(\mathbf{w}^{*\top}\mathbf{w}^0/2)\|\mathbf{v}^*\|_2^2 \geq (2+\Delta)\rho.$$

Therefore we have

$$\mathbf{v}^{*\top}\overline{\mathbf{v}}^{t+1} \geq (1 - \alpha\Delta)\rho + \alpha(2+\Delta)\rho - \alpha\rho = (1+\alpha)\rho.$$

Moreover,

$$|\mathbf{v}^{*\top}\mathbf{v}^{t+1} - \mathbf{v}^{*\top}\overline{\mathbf{v}}^{t+1}| \leq \alpha\|\mathbf{v}^*\|_2\eta_v.$$

By the assumptions on $n$ we have $\|\mathbf{v}^*\|_2\eta_v \leq \rho$. Therefore

$$\mathbf{v}^{*\top}\mathbf{v}^{t+1} \geq (1+\alpha)\rho - \alpha\|\mathbf{v}^*\|_2\eta_v \geq \rho.$$

Finally, we check $\kappa^2(\mathbf{1}^\top\mathbf{v}^*)\mathbf{1}^\top(\mathbf{v}^{t+1}-\mathbf{v}^*) \le \rho$. By definition we have

$$\mathbf{1}^\top\overline{\mathbf{g}}_v^t = \mathbf{1}^\top\{(\Delta\mathbf{I}+\kappa^2\mathbf{1}\mathbf{1}^\top)\mathbf{v}^t - [\phi(\mathbf{w}^t,\mathbf{w}^*)\mathbf{I}+\kappa^2\mathbf{1}\mathbf{1}^\top]\mathbf{v}^*\}$$
$$= (\Delta+\kappa^2 k)\mathbf{1}^\top\mathbf{v}^t - [\phi(\mathbf{w}^t,\mathbf{w}^*)+\kappa^2 k]\mathbf{1}^\top\mathbf{v}^*.$$

By Lemma 4.1, we have $\phi(\mathbf{w}^t,\mathbf{w}^*) \le \Delta$. Therefore,

$$(\mathbf{1}^\top\mathbf{v}^*)(\mathbf{1}^\top\overline{\mathbf{g}}_v^t) = (\Delta+\kappa^2 k)(\mathbf{1}^\top\mathbf{v}^*)(\mathbf{1}^\top\mathbf{v}^t) - [\phi(\mathbf{w}^t,\mathbf{w}^*)+\kappa^2 k](\mathbf{1}^\top\mathbf{v}^*)^2$$
$$\ge (\Delta+\kappa^2 k)(\mathbf{1}^\top\mathbf{v}^*)\mathbf{1}^\top(\mathbf{v}^t-\mathbf{v}^*),$$

and

$$(\mathbf{1}^\top\mathbf{v}^*)\mathbf{1}^\top(\overline{\mathbf{v}}^{t+1}-\mathbf{v}^*) \le (\mathbf{1}^\top\mathbf{v}^*)\mathbf{1}^\top(\mathbf{v}^t-\mathbf{v}^*) - \alpha(\Delta+\kappa^2 k)(\mathbf{1}^\top\mathbf{v}^*)\mathbf{1}^\top(\mathbf{v}^t-\mathbf{v}^*)$$
$$= [1-\alpha(\Delta+\kappa^2 k)](\mathbf{1}^\top\mathbf{v}^*)\mathbf{1}^\top(\mathbf{v}^t-\mathbf{v}^*).$$

Since $\mathbf{v}^t \in \mathcal{V}$, by Lemma 5.3, when $1-\alpha(\Delta+\kappa^2 k) \ge 0$ we have

$$\kappa^2(\mathbf{1}^\top\mathbf{v}^*)\mathbf{1}^\top(\mathbf{v}^{t+1}-\mathbf{v}^*) \le [1-\alpha(\Delta+\kappa^2 k)][\kappa^2(\mathbf{1}^\top\mathbf{v}^*)\mathbf{1}^\top(\mathbf{v}^t-\mathbf{v}^*)] + \alpha\kappa^2|\mathbf{1}^\top\mathbf{v}^*|\sqrt{k}\eta_v$$
$$\le [1-\alpha(\Delta+\kappa^2 k)]\rho + \alpha\kappa^2|\mathbf{1}^\top\mathbf{v}^*|\sqrt{k}\eta_v$$
$$\le [1-\alpha(\Delta+\kappa^2 k)]\rho + \alpha(\Delta+\kappa^2 k)\cdot\frac{\kappa^2|\mathbf{1}^\top\mathbf{v}^*|\sqrt{k}}{\Delta+\kappa^2 k}\cdot\eta_v$$
$$\le [1-\alpha(\Delta+\kappa^2 k)]\rho + \alpha(\Delta+\kappa^2 k)\cdot\frac{|\mathbf{1}^\top\mathbf{v}^*|}{\sqrt{k}}\cdot\eta_v.$$

By the assumption on $n$ we have $|\mathbf{1}^\top\mathbf{v}^*|k^{-1/2}\eta_v \le \rho$. Plugging it into the inequality above gives

$$\kappa^2(\mathbf{1}^\top\mathbf{v}^*)\mathbf{1}^\top(\mathbf{v}^{t+1}-\mathbf{v}^*) \le \rho.$$

Therefore we have $(\mathbf{w}^{t+1},\mathbf{v}^{t+1}) \in \mathcal{W}\times\mathcal{V}$. $\qquad\square$

## B.5 Proof of Lemma 5.5

The following auxiliary lemma plays a key role in converting the recursive bounds to explicit bounds.

**Lemma B.10.** Let $a,b \in (0,1)$, $c_1,c_2 > 0$ be constants. If sequences $\{u_t\}_{t\ge0}$, $\{v_t\}_{t\ge0}$ satisfies

$$u_{t+1} \le au_t + c_1 b^t + c_2,\ v_{t+1} \le av_t + c_1 t^2 b^t + c_2$$

then it holds that

$$u_t \le a^t u_0 + c_1 t(a\vee b)^{t-1} + c_2\frac{1}{1-a},\ v_t \le a^t v_0 + \frac{c_1}{3}t^3(a\vee b)^{t-1} + c_2\frac{1}{1-a}.$$

The proof of Lemma B.10 is given in Section E in appendix. We can now apply Lemma B.10 to the recursive bounds (5.7), (5.8) and (5.9).

*Proof of Lemma 5.5.* The first convergence result for $\mathbf{w}^t$ directly follows by Lemma 5.4 and (5.7). To prove (4.4), we first derive the convergence rate of $|\mathbf{1}^\top(\mathbf{v}^t-\mathbf{v}^*)|$. By (4.3) and Lemma 5.4, we have

$$|\mathbf{1}^\top(\mathbf{v}^{t+1}-\mathbf{v}^*)| \le \gamma_3|\mathbf{1}^\top(\mathbf{v}^t-\mathbf{v}^*)| + \alpha L|\mathbf{1}^\top\mathbf{v}^*|\gamma_1^t\|\mathbf{w}^0-\mathbf{w}^*\|_2 + 8\alpha L|\mathbf{1}^\top\mathbf{v}^*|\rho^{-1}\gamma_1\eta_w$$
$$+ \alpha\sqrt{k}\eta_v.$$

By Lemma B.10, we have

$$|\mathbf{1}^\top(\mathbf{v}^t-\mathbf{v}^*)| \le \gamma_3^t|\mathbf{1}^\top(\mathbf{v}^0-\mathbf{v}^*)| + t(\gamma_1\vee\gamma_3)^t\alpha L|\mathbf{1}^\top\mathbf{v}^*|\|\mathbf{w}^0-\mathbf{w}^*\|_2$$
$$+ \frac{8L|\mathbf{1}^\top\mathbf{v}^*|\rho^{-1}\gamma_1\eta_w + \sqrt{k}\eta_v}{\Delta+\kappa^2 k}.$$

Therefore, by Lemma 5.4 we have

$$\|\mathbf{v}^{t+1} - \mathbf{v}^*\|_2^2 \le \gamma_2^2 \|\mathbf{v}^t - \mathbf{v}^*\|_2^2 + 2(\Delta^{-1} + 4\alpha)\alpha L^2 \|\mathbf{v}^*\|_2^2 (\gamma_1^{2t}\|\mathbf{w}^0 - \mathbf{w}^*\|_2^2 + 64\rho^{-2}\gamma_1^{-4}\eta_w^2)$$

$$+ 12\alpha^2 \kappa^4 k \left\{ \gamma_3^{2t} |\mathbf{1}^\top(\mathbf{v}^0 - \mathbf{v}^*)|^2 + t^2(\gamma_1 \vee \gamma_3)^{2t}\alpha^2 L^2 |\mathbf{1}^\top \mathbf{v}^*|^2 \|\mathbf{w}^0 - \mathbf{w}^*\|_2^2 \right.$$

$$\left. + \left( \frac{8L|\mathbf{1}^\top \mathbf{v}^*|\rho^{-1}\gamma_1 \eta_w + \sqrt{k}\eta_v}{\Delta + \kappa^2 k} \right)^2 \right\}$$

$$+ \left( \frac{2\alpha}{\Delta} + 4\alpha^2 \right)\eta_v^2$$

$$\le \gamma_2^2 \|\mathbf{v}^t - \mathbf{v}^*\|_2^2 + R_1' t^2 (\gamma_1 \vee \gamma_3)^{2t} + R_2'.$$

where

$$R_1' = 2(\Delta^{-1} + 4\alpha)\alpha L^2 \|\mathbf{v}^*\|_2^2 \|\mathbf{w}^0 - \mathbf{w}^*\|_2^2 + 12\alpha^2 \kappa^4 k [|\mathbf{1}^\top(\mathbf{v}^0 - \mathbf{v}^*)|^2$$
$$+ \alpha^2 L^2 |\mathbf{1}^\top \mathbf{v}^*|^2 \|\mathbf{w}^0 - \mathbf{w}^*\|_2^2],$$

$$R_2' = 128(\Delta^{-1} + 4\alpha)\alpha L^2 \|\mathbf{v}^*\|_2^2 \rho^{-2}\gamma_1^{-4}\eta_w^2 + 12\alpha^2 \kappa^4 k \left( \frac{8L|\mathbf{1}^\top \mathbf{v}^*|\rho^{-1}\gamma_1 \eta_w + \sqrt{k}\eta_v}{\Delta + \kappa^2 k} \right)^2$$

$$+ \left( \frac{2\alpha}{\Delta} + 4\alpha^2 \right)\eta_v^2.$$

We can then further apply the second bound in Lemma B.10 to the above inequality and obtain

$$\|\mathbf{v}^t - \mathbf{v}^*\|_2 \le R_1 t^{3/2}(\gamma_1 \vee \gamma_2 \vee \gamma_3)^t + (R_2 + R_3 |\kappa|\sqrt{k})(\eta_w + \eta_v),$$

where

$$R_1 = \sqrt{2(\Delta^{-1} + 4\alpha)\alpha L^2}\|\mathbf{v}^*\|_2 \|\mathbf{w}^0 - \mathbf{w}^*\|_2 + 4\alpha\kappa^2\sqrt{k}[|\mathbf{1}^\top(\mathbf{v}^0 - \mathbf{v}^*)| + \|\mathbf{v}^0 - \mathbf{v}^*\|_2$$
$$+ \alpha L |\mathbf{1}^\top \mathbf{v}^*| \|\mathbf{w}^0 - \mathbf{w}^*\|_2],$$

$$R_2 = \frac{1}{\Delta\sqrt{1 - 4\alpha\Delta}} \left[ 16\sqrt{(1 + 4\alpha\Delta)L^2}\|\mathbf{v}^*\|_2 \rho^{-1}\gamma_1^{-2} + 32\sqrt{\alpha\Delta}\kappa L |\mathbf{1}^\top \mathbf{v}^*|\rho^{-1}\gamma_1 + \sqrt{2 + 4\alpha\Delta} \right],$$

$$R_3 = \sqrt{\frac{12\alpha}{\Delta(1 - 4\alpha\Delta)}}.$$

This completes the proof of convergence of $v^t$. $\qquad\square$

# C  Proofs of Lemmas in Appendix A

## C.1  Proof of Lemma A.1

The following lemma gives a generalized version of Hoeffding's Covariance Identity [21]. This result is mentioned in [31], and a version for bounded random variables is proved by [9]. We give the proof of the version we present in Appendix E.

**Lemma C.1.** Let $X_1$, $X_2$ be two continuous random variables. For right-continuous monotonic functions $f$ and $g$ we have

$$\mathrm{Cov}[f(X_1), g(X_2)] = \int_{\mathbb{R}^2} \mathrm{Cov}[\mathbb{1}(X_1 \le x_1), \mathbb{1}(X_2 \le x_2)]\mathrm{d}f(x_1)\mathrm{d}f(x_2).$$

We also need the following lemma, which is a Gaussian comparison inequality given by [34].

**Lemma C.2** ([34]). Let $\mathbf{X}, \mathbf{Y} \in \mathbb{R}^d$ be two centered Gaussian random vectors. If $\mathbb{E}(X_i^2) = \mathbb{E}(Y_i^2)$ and $\mathbb{E}(X_i X_j) \le \mathbb{E}(Y_i Y_j)$ for all $i, j = 1, \dots, d$, then for any real numbers $\tau_1, \dots, \tau_d$, we have

$$\mathbb{P}(X_1 \le \tau_1, \dots, X_d \le \tau_d) \le \mathbb{P}(Y_1 \le \tau_1, \dots, Y_d \le \tau_d).$$

*Proof of Lemma A.1.* It directly follows by Lemma C.1 and Lemma C.2 that $\mathrm{Cov}[f(Z_1), g(Z_2)]$ is an increasing function of $\theta$. Moreover, let $U_1$ and $U_2$ be independent standard Gaussian random variables. Then it is easy to check that

$$\frac{\mathrm{d}}{\mathrm{d}\theta}\mathbb{P}(U_1 \leq x_1, \theta U_1 + \sqrt{1-\theta^2}U_2 \leq x_2)\bigg|_{\theta=0} > 0$$

for all $x_1, x_2 \in \mathbb{R}$. Therefore for non-trivial increasing functions $f$ and $g$, by Lemma C.1 and the definition of Lebesgue-Stieltjes integration, we have $\mathrm{Cov}[f(Z_1), g(Z_2)] > 0$. □

# D  Proofs of Lemmas in Appendix B

## D.1  Proof of Lemma B.1

*Proof of Lemma B.1.* Let $\mathcal{N}_1 = \mathcal{N}(S^{r-1}, 1/8)$, $\mathcal{N}_2 = \mathcal{N}(S^{k-1}, 1/8)$ be $1/8$-nets covering $S^{r-1}$ and $S^{k-1}$ respectively. Then by Lemma 5.2 in [36] we have $|\mathcal{N}_1| \leq 17^r$ and $|\mathcal{N}_2| \leq 17^k$. Denote

$$A(\widehat{\mathbf{a}}, \widehat{\mathbf{a}}', \widehat{\mathbf{b}}, \widehat{\mathbf{b}}') = \frac{1}{n}\sum_{i=1}^{n}\sum_{j=1}^{k}|\widehat{b}_j\widehat{\mathbf{a}}^\top\mathbf{P}_j\mathbf{x}_i| \cdot \left|\sum_{j'=1}^{k}\widehat{b}'_{j'}\widehat{\mathbf{a}}'^\top\mathbf{P}_{j'}\mathbf{x}_i\right|.$$

It is easy to see that $\sum_{j'=1}^{k}\widehat{b}'_{j'}\widehat{\mathbf{a}}'^\top\mathbf{P}_{j'}\mathbf{x}_i$, $i = 1, \ldots, n$ are independent standard normal random variables. Moreover, for each $i = 1, \ldots, n$, $\widehat{\mathbf{a}}^\top\mathbf{P}_j\mathbf{x}_i$, $j = 1, \ldots, k$ are independent standard Gaussian random vectors. By triangle inequality we have

$$\left\|\sum_{j=1}^{k}|\widehat{b}_j\widehat{\mathbf{a}}^\top\mathbf{P}_j\mathbf{x}_i|\right\|_{\psi_2} \leq C_1\|\widehat{\mathbf{b}}\|_1 \leq C_1\sqrt{k},$$

where $C_1$ is an absolute constant. Therefore, by Lemma E.1 we have

$$\left\|\sum_{j=1}^{k}|\widehat{b}_j\widehat{\mathbf{a}}^\top\mathbf{P}_j\mathbf{x}_i| \cdot \left|\sum_{j'=1}^{k}\widehat{b}'_{j'}\widehat{\mathbf{a}}'^\top\mathbf{P}_{j'}\mathbf{x}_i\right|\right\|_{\psi_1} \leq C_2\sqrt{k},$$

where $C_2$ is an absolute constant. By Proposition 5.16 in [36], with probability at least $1 - \delta_1$ we have

$$|A(\widehat{\mathbf{a}}, \widehat{\mathbf{a}}', \widehat{\mathbf{b}}, \widehat{\mathbf{b}}') - \mathbb{E}[A(\widehat{\mathbf{a}}, \widehat{\mathbf{a}}', \widehat{\mathbf{b}}, \widehat{\mathbf{b}}')]| \leq C_3\sqrt{k}\sqrt{\frac{(r+k)\log(34/\delta_1)}{n}}$$

for all $\widehat{\mathbf{a}}, \widehat{\mathbf{a}}' \in \mathcal{N}_1$ and $\widehat{\mathbf{b}}, \widehat{\mathbf{b}}' \in \mathcal{N}_2$, where $C_3$ is an absolute constant. By the assumptions on $n$ we have

$$|A(\widehat{\mathbf{a}}, \widehat{\mathbf{a}}', \widehat{\mathbf{b}}, \widehat{\mathbf{b}}') - \mathbb{E}[A(\widehat{\mathbf{a}}, \widehat{\mathbf{a}}', \widehat{\mathbf{b}}, \widehat{\mathbf{b}}')]| \leq C_3\sqrt{k}.$$

Moreover, by the definition of $\psi_1$-norm we have

$$\mathbb{E}[A(\widehat{\mathbf{a}}, \widehat{\mathbf{a}}', \widehat{\mathbf{b}}, \widehat{\mathbf{b}}')] \leq \|A(\widehat{\mathbf{a}}, \widehat{\mathbf{a}}', \widehat{\mathbf{b}}, \widehat{\mathbf{b}}')\|_{\psi_1} \leq C_2\sqrt{k}.$$

Therefore

$$|A(\widehat{\mathbf{a}}, \widehat{\mathbf{a}}', \widehat{\mathbf{b}}, \widehat{\mathbf{b}}')| \leq C_4\sqrt{k}$$

for all $\widehat{\mathbf{a}}, \widehat{\mathbf{a}}' \in \mathcal{N}_1$ and $\widehat{\mathbf{b}}, \widehat{\mathbf{b}}' \in \mathcal{N}_2$, where $C_4$ is an absolute constant.

For any $\mathbf{a}, \mathbf{a}' \in S^{r-1}$ and $\mathbf{b}, \mathbf{b}' \in S^{k-1}$, there exists $\widehat{\mathbf{a}}, \widehat{\mathbf{a}}' \in \mathcal{N}_1$ and $\widehat{\mathbf{b}}, \widehat{\mathbf{b}}' \in \mathcal{N}_2$ such that

$$\|\mathbf{a} - \widehat{\mathbf{a}}\|_2, \|\mathbf{a}' - \widehat{\mathbf{a}}'\|_2, \|\mathbf{b} - \widehat{\mathbf{b}}\|_2, \|\mathbf{b}' - \widehat{\mathbf{b}}'\|_2 \leq 1/8.$$

Therefore

$$A(\mathbf{a}, \mathbf{a}', \mathbf{b}, \mathbf{b}') \leq C_4\sqrt{k} + I_1 + I_2 + I_3 + I_4, \tag{D.1}$$

where

$$I_1 = |A(\mathbf{a}, \mathbf{a}', \mathbf{b}, \mathbf{b}') - A(\mathbf{a}, \mathbf{a}', \mathbf{b}, \widehat{\mathbf{b}}')|,$$
$$I_2 = |A(\mathbf{a}, \mathbf{a}', \mathbf{b}, \widehat{\mathbf{b}}') - A(\mathbf{a}, \mathbf{a}', \widehat{\mathbf{b}}, \widehat{\mathbf{b}}')|,$$
$$I_3 = |A(\mathbf{a}, \mathbf{a}', \widehat{\mathbf{b}}, \widehat{\mathbf{b}}') - A(\mathbf{a}, \widehat{\mathbf{a}}', \widehat{\mathbf{b}}, \widehat{\mathbf{b}}')|,$$
$$I_4 = |A(\mathbf{a}, \widehat{\mathbf{a}}', \widehat{\mathbf{b}}, \widehat{\mathbf{b}}') - A(\widehat{\mathbf{a}}, \widehat{\mathbf{a}}', \widehat{\mathbf{b}}, \widehat{\mathbf{b}}')|.$$

Let

$$\overline{A} = \sup_{\substack{\mathbf{a}, \mathbf{a}' \in S^{r-1} \\ \mathbf{b}, \mathbf{b}' \in S^{k-1}}} A(\mathbf{a}, \mathbf{a}', \mathbf{b}, \mathbf{b}').$$

Then we have

$$I_1 \leq A(\mathbf{a}, \mathbf{a}', \mathbf{b}, \mathbf{b}' - \widehat{\mathbf{b}}')$$
$$= \|\mathbf{b}' - \widehat{\mathbf{b}}'\|_2 A\left(\mathbf{a}, \mathbf{a}', \mathbf{b}, \frac{\mathbf{b}' - \widehat{\mathbf{b}}'}{\|\mathbf{b}' - \widehat{\mathbf{b}}'\|_2}\right)$$
$$\leq \|\mathbf{b}' - \widehat{\mathbf{b}}'\|_2 \overline{A}.$$

Similarly, we have

$$I_2 \leq \|\mathbf{b} - \widehat{\mathbf{b}}\|_2 \overline{A},$$
$$I_3 \leq \|\mathbf{a}' - \widehat{\mathbf{a}}'\|_2 \overline{A},$$
$$I_4 \leq \|\mathbf{a} - \widehat{\mathbf{a}}\|_2 \overline{A}.$$

Plugging the inequalities above into (D.1) gives

$$\overline{A} \leq C_4 \sqrt{k} + \frac{1}{2}\overline{A}.$$

Therefore we have $\overline{A} \leq 2C_4\sqrt{k}$. This completes the proof. $\qquad\square$

## D.2 Proof of Lemma B.2

*Proof of Lemma B.2.* Let $\mathcal{N}_1 = \mathcal{N}(S^{r-1}, 1/8)$, $\mathcal{N}_2 = \mathcal{N}(S^{k-1}, 1/8)$ be $1/8$-nets covering $S^{r-1}$ and $S^{k-1}$ respectively. Then by Lemma 5.2 in [36] we have $|\mathcal{N}_1| \leq 17^r$ and $|\mathcal{N}_2| \leq 17^k$. Denote

$$A(\widehat{\mathbf{a}}, \widehat{\mathbf{a}}', \widehat{\mathbf{b}}, \widehat{\mathbf{b}}') = \frac{1}{n}\sum_{i=1}^{n}\sum_{j=1}^{k}\widehat{b}_j\sigma(\widehat{\mathbf{a}}^\top\mathbf{P}_j\mathbf{x}_i) \cdot \sum_{j'=1}^{k}\widehat{b}'_{j'}\widehat{\mathbf{a}}'^\top\mathbf{P}_{j'}\mathbf{x}_i.$$

For any $\widehat{\mathbf{a}}, \widehat{\mathbf{a}}' \in \mathcal{N}_1$ and $\widehat{\mathbf{b}}, \widehat{\mathbf{b}}' \in \mathcal{N}_2$, by Lemma B.8, $\sigma(\widehat{\mathbf{a}}^\top\mathbf{P}_j\mathbf{x}_i)$, $j = 1, \ldots, r$ are independent sub-Gaussian random variables with $\|\sigma(\widehat{\mathbf{a}}^\top\mathbf{P}_j\mathbf{x}_i)\|_{\psi_2} \leq C_1 L$ for some absolute constant $C_1$. Therefore by triangle inequality, we have

$$\left\|\sum_{j=1}^{k}\widehat{b}_j\sigma(\widehat{\mathbf{a}}^\top\mathbf{P}_j\mathbf{x}_i)\right\|_{\psi_2} \leq C_2 L\sqrt{k},$$

where $C_2$ is an absolute constant. Moreover, since $\mathbf{P}_{j'}\mathbf{x}_i$, $j' = 1, \ldots, k$ are independent Gaussian random vectors, we have

$$\left\|\sum_{j'=1}^{k}\widehat{b}'_{j'}\widehat{\mathbf{a}}'^\top\mathbf{P}_{j'}\mathbf{x}_i\right\|_{\psi_2} \leq C_3$$

for some absolute constant $C_3$. Therefore, by Lemma E.1 we have $\|A(\widehat{\mathbf{a}}, \widehat{\mathbf{a}}', \widehat{\mathbf{b}}, \widehat{\mathbf{b}}')\|_{\psi_1} \leq C_4 L\sqrt{k}$, where $C_4$ is an absolute constant. By Proposition 5.16 in [36], with probability at least $1 - \delta_2/2$, we have

$$|A(\widehat{\mathbf{a}}, \widehat{\mathbf{a}}', \widehat{\mathbf{b}}, \widehat{\mathbf{b}}') - \mathbb{E}A(\widehat{\mathbf{a}}, \widehat{\mathbf{a}}', \widehat{\mathbf{b}}, \widehat{\mathbf{b}}')| \leq C_5 L\sqrt{k}\sqrt{\frac{(r+k)\log(68/\delta_2)}{n}}$$

for all $\widehat{\mathbf{a}}, \widehat{\mathbf{a}}' \in \mathcal{N}_1$ and $\widehat{\mathbf{b}}, \widehat{\mathbf{b}}' \in \mathcal{N}_2$, where $C_5$ is an absolute constant. Therefore by the assumptions on $n$ we have

$$|A(\widehat{\mathbf{a}}, \widehat{\mathbf{a}}', \widehat{\mathbf{b}}, \widehat{\mathbf{b}}')| \leq C_6 L \sqrt{k} + |\mathbb{E}A(\widehat{\mathbf{a}}, \widehat{\mathbf{a}}', \widehat{\mathbf{b}}, \widehat{\mathbf{b}}')|,$$

where $C_6$ is an absolute constant. Moreover, by the definition of $\psi_1$-norm we have

$$\mathbb{E}A(\widehat{\mathbf{a}}, \widehat{\mathbf{a}}', \widehat{\mathbf{b}}, \widehat{\mathbf{b}}') \leq \|A(\widehat{\mathbf{a}}, \widehat{\mathbf{a}}', \widehat{\mathbf{b}}, \widehat{\mathbf{b}}')\|_{\psi_1} \leq C_4 L \sqrt{k}.$$

Therefore we have

$$|A(\widehat{\mathbf{a}}, \widehat{\mathbf{a}}', \widehat{\mathbf{b}}, \widehat{\mathbf{b}}')| \leq C_7 L \sqrt{k}$$

for some absolute constant $C_7$. Now for any $\mathbf{a}, \mathbf{a}' \in S^{r-1}$ and $\mathbf{b}, \mathbf{b}' \in S^{k-1}$, there exists $\widehat{\mathbf{a}}, \widehat{\mathbf{a}}' \in \mathcal{N}_1$ and $\widehat{\mathbf{b}}, \widehat{\mathbf{b}}' \in \mathcal{N}_2$ such that

$$\|\mathbf{a} - \widehat{\mathbf{a}}\|_2, \|\mathbf{a}' - \widehat{\mathbf{a}}'\|_2, \|\mathbf{b} - \widehat{\mathbf{b}}\|_2, \|\mathbf{b}' - \widehat{\mathbf{b}}'\|_2 \leq 1/8.$$

Therefore

$$A(\mathbf{a}, \mathbf{a}', \mathbf{b}, \mathbf{b}') \leq C_7 L \sqrt{k} + I_1 + I_2 + I_3 + I_4,$$

where

$$I_1 = |A(\mathbf{a}, \mathbf{a}', \mathbf{b}, \mathbf{b}') - A(\mathbf{a}, \mathbf{a}', \mathbf{b}, \widehat{\mathbf{b}}')|,$$
$$I_2 = |A(\mathbf{a}, \mathbf{a}', \mathbf{b}, \widehat{\mathbf{b}}') - A(\mathbf{a}, \mathbf{a}', \widehat{\mathbf{b}}, \widehat{\mathbf{b}}')|,$$
$$I_3 = |A(\mathbf{a}, \mathbf{a}', \widehat{\mathbf{b}}, \widehat{\mathbf{b}}') - A(\mathbf{a}, \widehat{\mathbf{a}}', \widehat{\mathbf{b}}, \widehat{\mathbf{b}}')|,$$
$$I_4 = |A(\mathbf{a}, \widehat{\mathbf{a}}', \widehat{\mathbf{b}}, \widehat{\mathbf{b}}') - A(\widehat{\mathbf{a}}, \widehat{\mathbf{a}}', \widehat{\mathbf{b}}, \widehat{\mathbf{b}}')|.$$

Let

$$\overline{A} = \sup_{\substack{\mathbf{a}, \mathbf{a}' \in S^{r-1} \\ \mathbf{b}, \mathbf{b}' \in S^{k-1}}} A(\mathbf{a}, \mathbf{a}', \mathbf{b}, \mathbf{b}').$$

Since $A(\mathbf{a}, \mathbf{a}', \mathbf{b}, \mathbf{b}')$ is linear in $\mathbf{a}', \mathbf{b}$ and $\mathbf{b}'$, we have

$$I_1 \leq \|\mathbf{b}' - \widehat{\mathbf{b}}'\|_2 \overline{A}, \ I_2 \leq \|\mathbf{b} - \widehat{\mathbf{b}}\|_2 \overline{A}, \ I_3 \leq \|\mathbf{a}' - \widehat{\mathbf{a}}'\|_2 \overline{A}$$

Moreover, by the Lipschitz continuity of $\sigma(\cdot)$, we have

$$I_4 \leq \frac{1}{n} \sum_{i=1}^{n} \sum_{j=1}^{k} |\widehat{b}_j (\mathbf{a} - \widehat{\mathbf{a}})^\top \mathbf{P}_j \mathbf{x}_i| \cdot \left| \sum_{j'=1}^{k} \widehat{b}'_{j'} \widehat{\mathbf{a}}'^\top \mathbf{P}_{j'} \mathbf{x}_i \right|.$$

Therefore by Lemma B.1 with $\delta_1 = \delta_2/2$ we have

$$I_4 \leq C_8 \sqrt{k} \|\mathbf{a} - \mathbf{a}'\|_2$$

for some absolute constant $C_8$. Summing the bounds on $I_1, \ldots, I_4$ gives

$$\overline{A} \leq C_9 L \sqrt{k} + \overline{A}/2,$$

where $C_9$ is an absolute constant. Therefore we have $\overline{A} \leq 2 C_9 L \sqrt{k}$. This completes the proof. $\quad\square$

### D.3 Proof of Lemma B.3

*Proof of Lemma B.3.* Let $\mathcal{N}_1 = \mathcal{N}(S^{r-1}, 1/8)$, $\mathcal{N}_2 = \mathcal{N}(S^{k-1}, 1/8)$ be $1/8$-nets covering $S^{r-1}$ and $S^{k-1}$ respectively. Then by Lemma 5.2 in [36] we have $|\mathcal{N}_1| \leq 17^r$ and $|\mathcal{N}_2| \leq 17^k$. Denote

$$A(\widehat{\mathbf{a}}, \widehat{\mathbf{a}}', \widehat{\mathbf{b}}, \widehat{\mathbf{b}}') = \frac{1}{n} \sum_{i=1}^{n} \sum_{j=1}^{k} |\widehat{b}_j \widehat{\mathbf{a}}^\top \mathbf{P}_j \mathbf{x}_i| \cdot \sum_{j'=1}^{k} |\widehat{b}'_{j'} \widehat{\mathbf{a}}'^\top \mathbf{P}_{j'} \mathbf{x}_i|.$$

For any $\widehat{\mathbf{a}}, \widehat{\mathbf{a}}' \in \mathcal{N}_1$ and $\widehat{\mathbf{b}}, \widehat{\mathbf{b}}' \in \mathcal{N}_2$, by triangle inequality, we have

$$\left\| \sum_{j=1}^{k} \widehat{b}_{j'} \widehat{\mathbf{a}}^\top \mathbf{P}_j \mathbf{x}_i \right\|_{\psi_2}, \left\| \sum_{j'=1}^{k} \widehat{b}_{j'} \widehat{\mathbf{a}}'^\top \mathbf{P}_{j'} \mathbf{x}_i \right\|_{\psi_2} \leq C_1 \sqrt{k}$$

for some absolute constant $C_1$. Therefore, by Lemma E.1 we have $\|A(\widehat{\mathbf{a}}, \widehat{\mathbf{a}}', \widehat{\mathbf{b}}, \widehat{\mathbf{b}}')\|_{\psi_1} \leq C_2 k$, where $C_2$ is an absolute constant. By Proposition 5.16 in [36], with probability at least $1 - \delta_3$, we have

$$|A(\widehat{\mathbf{a}}, \widehat{\mathbf{a}}', \widehat{\mathbf{b}}, \widehat{\mathbf{b}}') - \mathbb{E}[A(\widehat{\mathbf{a}}, \widehat{\mathbf{a}}', \widehat{\mathbf{b}}, \widehat{\mathbf{b}}')]| \leq C_3 k \sqrt{\frac{(r+k)\log(34/\delta_3)}{n}}$$

for all $\widehat{\mathbf{a}}, \widehat{\mathbf{a}}' \in \mathcal{N}_1$ and $\widehat{\mathbf{b}}, \widehat{\mathbf{b}}' \in \mathcal{N}_2$, where $C_3$ is an absolute constant. Therefore by the assumptions on $n$ we have

$$|A(\widehat{\mathbf{a}}, \widehat{\mathbf{a}}', \widehat{\mathbf{b}}, \widehat{\mathbf{b}}')| \leq C_3 k + |\mathbb{E}[A(\widehat{\mathbf{a}}, \widehat{\mathbf{a}}', \widehat{\mathbf{b}}, \widehat{\mathbf{b}}')]|.$$

By the definition of $\psi_1$-norm we have

$$\mathbb{E}[A(\widehat{\mathbf{a}}, \widehat{\mathbf{a}}', \widehat{\mathbf{b}}, \widehat{\mathbf{b}}')] \leq \|A(\widehat{\mathbf{a}}, \widehat{\mathbf{a}}', \widehat{\mathbf{b}}, \widehat{\mathbf{b}}')\|_{\psi_1} \leq C_2 k.$$

Therefore we have

$$|A(\widehat{\mathbf{a}}, \widehat{\mathbf{a}}', \widehat{\mathbf{b}}, \widehat{\mathbf{b}}')| \leq C_4 k,$$

where $C_4$ is an absolute constant. Now for any $\mathbf{a}, \mathbf{a}' \in S^{r-1}$ and $\mathbf{b}, \mathbf{b}' \in S^{k-1}$, there exists $\widehat{\mathbf{a}}, \widehat{\mathbf{a}}' \in \mathcal{N}_1$ and $\widehat{\mathbf{b}}, \widehat{\mathbf{b}}' \in \mathcal{N}_2$ such that

$$\|\mathbf{a} - \widehat{\mathbf{a}}\|_2, \|\mathbf{a}' - \widehat{\mathbf{a}}'\|_2, \|\mathbf{b} - \widehat{\mathbf{b}}\|_2, \|\mathbf{b}' - \widehat{\mathbf{b}}'\|_2 \leq 1/8.$$

Therefore

$$A(\mathbf{a}, \mathbf{a}', \mathbf{b}, \mathbf{b}') \leq C_4 k + I_1 + I_2 + I_3 + I_4,$$

where

$$I_1 = |A(\mathbf{a}, \mathbf{a}', \mathbf{b}, \mathbf{b}') - A(\mathbf{a}, \mathbf{a}', \mathbf{b}, \widehat{\mathbf{b}}')|,$$
$$I_2 = |A(\mathbf{a}, \mathbf{a}', \mathbf{b}, \widehat{\mathbf{b}}') - A(\mathbf{a}, \mathbf{a}', \widehat{\mathbf{b}}, \widehat{\mathbf{b}}')|,$$
$$I_3 = |A(\mathbf{a}, \mathbf{a}', \widehat{\mathbf{b}}, \widehat{\mathbf{b}}') - A(\mathbf{a}, \widehat{\mathbf{a}}', \widehat{\mathbf{b}}, \widehat{\mathbf{b}}')|,$$
$$I_4 = |A(\mathbf{a}, \widehat{\mathbf{a}}', \widehat{\mathbf{b}}, \widehat{\mathbf{b}}') - A(\widehat{\mathbf{a}}, \widehat{\mathbf{a}}', \widehat{\mathbf{b}}, \widehat{\mathbf{b}}')|.$$

Let

$$\overline{A} = \sup_{\substack{\mathbf{a}, \mathbf{a}' \in S^{r-1} \\ \mathbf{b}, \mathbf{b}' \in S^{k-1}}} A(\mathbf{a}, \mathbf{a}', \mathbf{b}, \mathbf{b}').$$

Since $A(\mathbf{a}, \mathbf{a}', \mathbf{b}, \mathbf{b}')$ is Lipschitz continuous in $\mathbf{a}, \mathbf{a}', \mathbf{b}$ and $\mathbf{b}'$, we have

$$I_1 \leq \|\mathbf{b}' - \widehat{\mathbf{b}}'\|_2 \overline{A}, \ I_2 \leq \|\mathbf{b} - \widehat{\mathbf{b}}\|_2 \overline{A}, \ I_3 \leq \|\mathbf{a}' - \widehat{\mathbf{a}}'\|_2 \overline{A}, \ I_4 \leq \|\mathbf{a} - \widehat{\mathbf{a}}\|_2 \overline{A}.$$

Therefore,

$$\overline{A} \leq C_4 k + \overline{A}/2,$$

Therefore we have $\overline{A} \leq 2C_4 k$. This completes the proof. $\qquad \square$

### D.4 Proof of Lemma B.4

*Proof of Lemma B.4.* Let $\mathcal{N}_1 = \mathcal{N}(S^{r-1}, 1/8)$, $\mathcal{N}_2 = \mathcal{N}(S^{k-1}, 1/8)$ be $1/8$-nets covering $S^{r-1}$ and $S^{k-1}$ respectively. Then by Lemma 5.2 in [36] we have $|\mathcal{N}_1| \leq 17^r$ and $|\mathcal{N}_2| \leq 17^k$. Denote

$$A(\widehat{\mathbf{a}}, \widehat{\mathbf{a}}', \widehat{\mathbf{b}}, \widehat{\mathbf{b}}') = \frac{1}{n} \sum_{i=1}^{n} \sum_{j=1}^{k} |\widehat{b}_j \widehat{\mathbf{a}}^\top \mathbf{P}_j \mathbf{x}_i| \cdot \sum_{j'=1}^{k} |\widehat{b}_{j'} \sigma('\widehat{\mathbf{a}}'^\top \mathbf{P}_{j'} \mathbf{x}_i)|.$$

For any $\widehat{\mathbf{a}}, \widehat{\mathbf{a}}' \in \mathcal{N}_1$ and $\widehat{\mathbf{b}}, \widehat{\mathbf{b}}' \in \mathcal{N}_2$, by Lemma B.8, $\sigma(\widehat{\mathbf{a}}'^\top \mathbf{P}_{j'} \mathbf{x}_i)$, $j' = 1, \dots, r$ are independent sub-Gaussian random variables with $\|\sigma(\widehat{\mathbf{a}}'^\top \mathbf{P}_j \mathbf{x}_i)\|_{\psi_2} \leq C_1 L$ for some absolute constant $C_1$. Therefore by triangle inequality, we have

$$\left\| \sum_{j=1}^k |\widehat{b}_j \sigma(\widehat{\mathbf{a}}^\top \mathbf{P}_j \mathbf{x}_i)| \right\|_{\psi_2} \leq C_2 L \sqrt{k},$$

where $C_2$ is an absolute constant. Similarly, we have

$$\left\| \sum_{j'=1}^k |\widehat{b}'_{j'} \widehat{\mathbf{a}}'^\top \mathbf{P}_{j'} \mathbf{x}_i| \right\|_{\psi_2} \leq C_3 \sqrt{k}$$

for some absolute constant $C_3$. Therefore, by Lemma E.1 we have $\|A(\widehat{\mathbf{a}}, \widehat{\mathbf{a}}', \widehat{\mathbf{b}}, \widehat{\mathbf{b}}')\|_{\psi_1} \leq C_4 Lk$, where $C_4$ is an absolute constant. By Proposition 5.16 in [36], with probability at least $1 - \delta_4/2$, we have

$$|A(\widehat{\mathbf{a}}, \widehat{\mathbf{a}}', \widehat{\mathbf{b}}, \widehat{\mathbf{b}}') - \mathbb{E}[A(\widehat{\mathbf{a}}, \widehat{\mathbf{a}}', \widehat{\mathbf{b}}, \widehat{\mathbf{b}}')]| \leq C_5 Lk \sqrt{\frac{(r+k)\log(68/\delta_4)}{n}}$$

for all $\widehat{\mathbf{a}}, \widehat{\mathbf{a}}' \in \mathcal{N}_1$ and $\widehat{\mathbf{b}}, \widehat{\mathbf{b}}' \in \mathcal{N}_2$, where $C_5$ is an absolute constant. Therefore by the assumptions on $n$ we have

$$|A(\widehat{\mathbf{a}}, \widehat{\mathbf{a}}', \widehat{\mathbf{b}}, \widehat{\mathbf{b}}')| \leq C_5 Lk + |\mathbb{E}[A(\widehat{\mathbf{a}}, \widehat{\mathbf{a}}', \widehat{\mathbf{b}}, \widehat{\mathbf{b}}')]|.$$

By the definition of $\psi_1$-norm we have

$$\mathbb{E}[A(\widehat{\mathbf{a}}, \widehat{\mathbf{a}}', \widehat{\mathbf{b}}, \widehat{\mathbf{b}}')] \leq \|A(\widehat{\mathbf{a}}, \widehat{\mathbf{a}}', \widehat{\mathbf{b}}, \widehat{\mathbf{b}}')\|_{\psi_1} \leq C_4 Lk.$$

Therefore we have

$$|A(\widehat{\mathbf{a}}, \widehat{\mathbf{a}}', \widehat{\mathbf{b}}, \widehat{\mathbf{b}}')| \leq C_6 Lk$$

for some absolute constant $C_6$. Now for any $\mathbf{a}, \mathbf{a}' \in S^{r-1}$ and $\mathbf{b}, \mathbf{b}' \in S^{k-1}$, there exists $\widehat{\mathbf{a}}, \widehat{\mathbf{a}}' \in \mathcal{N}_1$ and $\widehat{\mathbf{b}}, \widehat{\mathbf{b}}' \in \mathcal{N}_2$ such that

$$\|\mathbf{a} - \widehat{\mathbf{a}}\|_2, \|\mathbf{a}' - \widehat{\mathbf{a}}'\|_2, \|\mathbf{b} - \widehat{\mathbf{b}}\|_2, \|\mathbf{b}' - \widehat{\mathbf{b}}'\|_2 \leq 1/8.$$

Therefore

$$A(\mathbf{a}, \mathbf{a}', \mathbf{b}, \mathbf{b}') \leq C_6 Lk + I_1 + I_2 + I_3 + I_4,$$

where

$$I_1 = |A(\mathbf{a}, \mathbf{a}', \mathbf{b}, \mathbf{b}') - A(\mathbf{a}, \mathbf{a}', \mathbf{b}, \widehat{\mathbf{b}}')|,$$
$$I_2 = |A(\mathbf{a}, \mathbf{a}', \mathbf{b}, \widehat{\mathbf{b}}') - A(\mathbf{a}, \mathbf{a}', \widehat{\mathbf{b}}, \widehat{\mathbf{b}}')|,$$
$$I_3 = |A(\mathbf{a}, \mathbf{a}', \widehat{\mathbf{b}}, \widehat{\mathbf{b}}') - A(\mathbf{a}, \widehat{\mathbf{a}}', \widehat{\mathbf{b}}, \widehat{\mathbf{b}}')|,$$
$$I_4 = |A(\mathbf{a}, \widehat{\mathbf{a}}', \widehat{\mathbf{b}}, \widehat{\mathbf{b}}') - A(\widehat{\mathbf{a}}, \widehat{\mathbf{a}}', \widehat{\mathbf{b}}, \widehat{\mathbf{b}}')|.$$

Let

$$\overline{A} = \sup_{\substack{\mathbf{a}, \mathbf{a}' \in S^{r-1} \\ \mathbf{b}, \mathbf{b}' \in S^{k-1}}} A(\mathbf{a}, \mathbf{a}', \mathbf{b}, \mathbf{b}').$$

Since $A(\mathbf{a}, \mathbf{a}', \mathbf{b}, \mathbf{b}')$ is Lipschitz continuous in $\mathbf{a}$, $\mathbf{b}$ and $\mathbf{b}'$, we have

$$I_1 \leq \|\mathbf{b}' - \widehat{\mathbf{b}}'\|_2 \overline{A}, \ I_2 \leq \|\mathbf{b} - \widehat{\mathbf{b}}\|_2 \overline{A}, \ I_4 \leq \|\mathbf{a} - \widehat{\mathbf{a}}\|_2 \overline{A}.$$

Moreover, by the Lipschitz continuity of $\sigma(\cdot)$, we have

$$I_3 \leq \frac{1}{n} \sum_{i=1}^n \sum_{j=1}^k |\widehat{b}_j \mathbf{a}^\top \mathbf{P}_j \mathbf{x}_i| \cdot \sum_{j'=1}^k |\widehat{b}'_{j'} (\mathbf{a}' - \widehat{\mathbf{a}}')^\top \mathbf{P}_{j'} \mathbf{x}_i|.$$

Therefore by Lemma B.3 with $\delta_3 = \delta_4/2$ we have

$$I_3 \leq C_7 k \|\mathbf{a}' - \widehat{\mathbf{a}}'\|_2$$

for some absolute constant $C_7$. Since $L \geq 1$, summing the bounds on $I_1, \dots, I_4$ gives

$$\overline{A} \leq C_8 Lk + \overline{A}/2,$$

where $C_8$ is an absolute constant. Therefore we have $\overline{A} \leq 2C_8 Lk$. This completes the proof. $\qquad\square$

## D.5 Proof of Lemma B.5

*Proof of Lemma B.5.* Let $\mathcal{N}_1 = \mathcal{N}(S^{r-1}, 1/8)$, $\mathcal{N}_2 = \mathcal{N}(S^{k-1}, 1/8)$ be 1/8-nets covering $S^{r-1}$ and $S^{k-1}$ respectively. Then by Lemma 5.2 in [36] we have $|\mathcal{N}_1| \le 17^r$ and $|\mathcal{N}_2| \le 17^k$. Denote

$$A(\widehat{\mathbf{a}}, \widehat{\mathbf{a}}', \widehat{\mathbf{b}}, \widehat{\mathbf{b}}') = \frac{1}{n} \sum_{i=1}^{n} \sum_{j=1}^{k} \widehat{b}_j \sigma(\widehat{\mathbf{a}}^\top \mathbf{P}_j \mathbf{x}_i) \cdot \sum_{j'=1}^{k} \widehat{b}'_{j'} \sigma('\widehat{\mathbf{a}}'^\top \mathbf{P}_{j'} \mathbf{x}_i).$$

For any $\widehat{\mathbf{a}}, \widehat{\mathbf{a}}' \in \mathcal{N}_1$ and $\widehat{\mathbf{b}}, \widehat{\mathbf{b}}' \in \mathcal{N}_2$, similar to the proof of Lemma B.4, we have

$$\left\| \sum_{j=1}^{k} \widehat{b}_j \sigma(\widehat{\mathbf{a}}^\top \mathbf{P}_j \mathbf{x}_i) \right\|_{\psi_2}, \left\| \sum_{j'=1}^{k} \widehat{b}'_{j'} \sigma(\widehat{\mathbf{a}}'^\top \mathbf{P}_{j'} \mathbf{x}_i) \right\|_{\psi_2} \le C_1 L \sqrt{k},$$

where $C_1$ is an absolute constant. Therefore, by Lemma E.1 we have $\|A(\widehat{\mathbf{a}}, \widehat{\mathbf{a}}', \widehat{\mathbf{b}}, \widehat{\mathbf{b}}')\|_{\psi_1} \le C_2 L^2 k$, where $C_2$ is an absolute constant. By Proposition 5.16 in [36], with probability at least $1 - \delta_5/3$, we have

$$|A(\widehat{\mathbf{a}}, \widehat{\mathbf{a}}', \widehat{\mathbf{b}}, \widehat{\mathbf{b}}') - \mathbb{E}[A(\widehat{\mathbf{a}}, \widehat{\mathbf{a}}', \widehat{\mathbf{b}}, \widehat{\mathbf{b}}')]| \le C_3 L^2 k \sqrt{\frac{(r+k)\log(102/\delta_5)}{n}}$$

for all $\widehat{\mathbf{a}}, \widehat{\mathbf{a}}' \in \mathcal{N}_1$ and $\widehat{\mathbf{b}}, \widehat{\mathbf{b}}' \in \mathcal{N}_2$, where $C_3$ is an absolute constant. Therefore by the assumptions on $n$ we have

$$|A(\widehat{\mathbf{a}}, \widehat{\mathbf{a}}', \widehat{\mathbf{b}}, \widehat{\mathbf{b}}')| \le C_3 L^2 k + |\mathbb{E}[A(\widehat{\mathbf{a}}, \widehat{\mathbf{a}}', \widehat{\mathbf{b}}, \widehat{\mathbf{b}}')]|.$$

Similar to the proofs of Lemma B.1-B.4, by the definition of $\psi_1$-norm we have

$$\mathbb{E}[A(\widehat{\mathbf{a}}, \widehat{\mathbf{a}}', \widehat{\mathbf{b}}, \widehat{\mathbf{b}}')] \le \|A(\widehat{\mathbf{a}}, \widehat{\mathbf{a}}', \widehat{\mathbf{b}}, \widehat{\mathbf{b}}')\|_{\psi_1} \le C_2 L^2 k.$$

Therefore we have

$$|A(\widehat{\mathbf{a}}, \widehat{\mathbf{a}}', \widehat{\mathbf{b}}, \widehat{\mathbf{b}}')| \le C_4 L^2 k$$

for some absolute constant $C_4$. Now for any $\mathbf{a}, \mathbf{a}' \in S^{r-1}$ and $\mathbf{b}, \mathbf{b}' \in S^{k-1}$, there exists $\widehat{\mathbf{a}}, \widehat{\mathbf{a}}' \in \mathcal{N}_1$ and $\widehat{\mathbf{b}}, \widehat{\mathbf{b}}' \in \mathcal{N}_2$ such that

$$\|\mathbf{a} - \widehat{\mathbf{a}}\|_2, \|\mathbf{a}' - \widehat{\mathbf{a}}'\|_2, \|\mathbf{b} - \widehat{\mathbf{b}}\|_2, \|\mathbf{b}' - \widehat{\mathbf{b}}'\|_2 \le 1/8.$$

Therefore

$$A(\mathbf{a}, \mathbf{a}', \mathbf{b}, \mathbf{b}') \le C_4 L^2 k + I_1 + I_2 + I_3 + I_4,$$

where

$$I_1 = |A(\mathbf{a}, \mathbf{a}', \mathbf{b}, \mathbf{b}') - A(\mathbf{a}, \mathbf{a}', \mathbf{b}, \widehat{\mathbf{b}}')|,$$
$$I_2 = |A(\mathbf{a}, \mathbf{a}', \mathbf{b}, \widehat{\mathbf{b}}') - A(\mathbf{a}, \mathbf{a}', \widehat{\mathbf{b}}, \widehat{\mathbf{b}}')|,$$
$$I_3 = |A(\mathbf{a}, \mathbf{a}', \widehat{\mathbf{b}}, \widehat{\mathbf{b}}') - A(\mathbf{a}, \widehat{\mathbf{a}}', \widehat{\mathbf{b}}, \widehat{\mathbf{b}}')|,$$
$$I_4 = |A(\mathbf{a}, \widehat{\mathbf{a}}', \widehat{\mathbf{b}}, \widehat{\mathbf{b}}') - A(\widehat{\mathbf{a}}, \widehat{\mathbf{a}}', \widehat{\mathbf{b}}, \widehat{\mathbf{b}}')|.$$

Let

$$\overline{A} = \sup_{\substack{\mathbf{a}, \mathbf{a}' \in S^{r-1} \\ \mathbf{b}, \mathbf{b}' \in S^{k-1}}} A(\mathbf{a}, \mathbf{a}', \mathbf{b}, \mathbf{b}').$$

Since $A(\mathbf{a}, \mathbf{a}', \mathbf{b}, \mathbf{b}')$ is linear in $\mathbf{b}$ and $\mathbf{b}'$, we have

$$I_1 \le \|\mathbf{b}' - \widehat{\mathbf{b}}'\|_2 \overline{A}, \ I_2 \le \|\mathbf{b} - \widehat{\mathbf{b}}\|_2 \overline{A}.$$

Moreover, by the Lipschitz continuity of $\sigma(\cdot)$, we have

$$I_3 \le \frac{1}{n} \sum_{i=1}^{n} \sum_{j=1}^{k} |\widehat{b}_j \sigma(\mathbf{a}^\top \mathbf{P}_j \mathbf{x}_i)| \cdot \sum_{j'=1}^{k} |\widehat{b}'_{j'} (\mathbf{a}' - \widehat{\mathbf{a}}')^\top \mathbf{P}_{j'} \mathbf{x}_i|,$$

$$I_4 \le \frac{1}{n} \sum_{i=1}^{n} \sum_{j=1}^{k} |\widehat{b}_j (\mathbf{a} - \widehat{\mathbf{a}})^\top \mathbf{P}_j \mathbf{x}_i| \cdot \sum_{j'=1}^{k} |\widehat{b}'_{j'} \sigma(\widehat{\mathbf{a}}'^\top \mathbf{P}_{j'} \mathbf{x}_i)|,$$

Therefore by Lemma B.4 with $\delta_4 = 2\delta_5/3$ we have

$$I_3 \leq C_5 k \|\mathbf{a}' - \widehat{\mathbf{a}}'\|_2, \; I_4 \leq C_5 k \|\mathbf{a} - \widehat{\mathbf{a}}\|_2$$

for some absolute constant $C_5$. Since we have $L \geq 1$, summing the bounds on $I_1, \ldots, I_4$ gives

$$\overline{A} \leq C_6 L^2 k + \overline{A}/2,$$

where $C_6$ is an absolute constant. Therefore we have $\overline{A} \leq 2C_6 L^2 k$. This completes the proof. □

## D.6 Proof of Lemma B.6

*Proof of Lemma B.6.* Let $\mathcal{N}_1 = \mathcal{N}(S^{r-1}, 1/4)$, $\mathcal{N}_2 = \mathcal{N}(S^{k-1}, 1/4)$ be $1/4$-nets covering $S^{r-1}$ and $S^{k-1}$ respectively. Then by Lemma 5.2 in [36] we have $|\mathcal{N}_1| \leq 9^r$ and $|\mathcal{N}_2| \leq 9^k$. Denote

$$A(\widehat{\mathbf{a}}, \widehat{\mathbf{b}}) = \frac{1}{n} \sum_{i=1}^n \epsilon_i \cdot \sum_{j=1}^k \widehat{b}_j \widehat{\mathbf{a}}^\top \mathbf{P}_j \mathbf{x}_i.$$

For any $\widehat{\mathbf{a}} \in \mathcal{N}_1$ and $\widehat{\mathbf{b}} \in \mathcal{N}_2$, since $\mathbf{P}_j \mathbf{x}_i$, $j = 1, \ldots, k$ are independent Gaussian random vectors, we have

$$\left\| \sum_{j=1}^k \widehat{b}_j \widehat{\mathbf{a}}^\top \mathbf{P}_j \mathbf{x}_i \right\|_{\psi_2} \leq C_1$$

for some absolute constant $C_1$. Therefore by Lemma E.1 we have $\|A(\widehat{\mathbf{a}}, \widehat{\mathbf{b}})\|_{\psi_1} \leq C_2 \nu$, where $C_2$ is an absolute constant. Since $\mathbb{E} A(\widehat{\mathbf{a}}, \widehat{\mathbf{b}}) = 0$, by Proposition 5.16 in [36], with probability at least $1 - \delta_6$, we have

$$|A(\widehat{\mathbf{a}}, \widehat{\mathbf{b}})| \leq C_3 \nu \sqrt{\frac{(r + k) \log(18/\delta_6)}{n}}$$

for all $\widehat{\mathbf{a}} \in \mathcal{N}_1$ and $\widehat{\mathbf{b}} \in \mathcal{N}_2$, where $C_3$ is an absolute constant. Therefore by the assumptions on $n$ we have

$$|A(\widehat{\mathbf{a}}, \widehat{\mathbf{b}})| \leq C_3 \nu.$$

Now for any $\mathbf{a} \in S^{r-1}$ and $\mathbf{b} \in S^{k-1}$, there exists $\widehat{\mathbf{a}} \in \mathcal{N}_1$ and $\widehat{\mathbf{b}} \in \mathcal{N}_2$ such that

$$\|\mathbf{a} - \widehat{\mathbf{a}}\|_2, \|\mathbf{b} - \widehat{\mathbf{b}}\|_2 \leq 1/4.$$

Therefore

$$A(\mathbf{a}, \mathbf{b}) \leq C_3 \nu + I_1 + I_2, \tag{D.2}$$

where

$$I_1 = |A(\mathbf{a}, \mathbf{b}) - A(\mathbf{a}, \widehat{\mathbf{b}})|, \; I_2 = |A(\mathbf{a}, \widehat{\mathbf{b}}) - A(\widehat{\mathbf{a}}, \widehat{\mathbf{b}})|.$$

Let

$$\overline{A} = \sup_{\substack{\mathbf{a} \in S^{r-1} \\ \mathbf{b} \in S^{k-1}}} A(\mathbf{a}, \mathbf{b}).$$

Since $A(\mathbf{a}, \mathbf{b})$ is linear in $\mathbf{a}$ and $\mathbf{b}$, we have

$$I_1 \leq \|\mathbf{b} - \widehat{\mathbf{b}}\|_2 \overline{A}, \; I_2 \leq \|\mathbf{a} - \widehat{\mathbf{a}}\|_2 \overline{A}.$$

Plugging the two inequalities above into (D.2) gives

$$\overline{A} \leq C_3 \nu + \overline{A}/2.$$

Therefore we have $\overline{A} \leq 2C_3 \nu$. This completes the proof. □

## D.7  Proof of Lemma B.7

*Proof of Lemma B.7.* Let $\mathcal{N}_1 = \mathcal{N}(S^{r-1}, 1/4)$, $\mathcal{N}_2 = \mathcal{N}(S^{k-1}, 1/4)$ be $1/4$-nets covering $S^{r-1}$ and $S^{k-1}$ respectively. Then by Lemma 5.2 in [36] we have $|\mathcal{N}_1| \leq 9^r$ and $|\mathcal{N}_2| \leq 9^k$. Denote

$$A(\widehat{\mathbf{a}}, \widehat{\mathbf{b}}) = \frac{1}{n} \sum_{i=1}^{n} |\epsilon_i| \cdot \sum_{j=1}^{k} |\widehat{b}_j \widehat{\mathbf{a}}^\top \mathbf{P}_j \mathbf{x}_i|.$$

For any $\widehat{\mathbf{a}} \in \mathcal{N}_1$ and $\widehat{\mathbf{b}} \in \mathcal{N}_2$, since $\mathbf{P}_j \mathbf{x}_i$, $j = 1, \ldots, k$ are independent Gaussian random vectors, by triangle inequality we have

$$\left\| \sum_{j=1}^{k} |\widehat{b}_j \widehat{\mathbf{a}}^\top \mathbf{P}_j \mathbf{x}_i| \right\|_{\psi_2} \leq C_1 \sqrt{k}$$

for some absolute constant $C_1$. Therefore by Lemma E.1 we have $\|A(\widehat{\mathbf{a}}, \widehat{\mathbf{b}})\|_{\psi_1} \leq C_2 \nu \sqrt{k}$, where $C_2$ is an absolute constant. By Proposition 5.16 in [36], with probability at least $1 - \delta_7$, we have

$$|A(\widehat{\mathbf{a}}, \widehat{\mathbf{b}}) - \mathbb{E}[A(\widehat{\mathbf{a}}, \widehat{\mathbf{b}})]| \leq C_3 \nu \sqrt{k} \sqrt{\frac{(r+k)\log(18/\delta_7)}{n}}$$

for all $\widehat{\mathbf{a}} \in \mathcal{N}_1$ and $\widehat{\mathbf{b}} \in \mathcal{N}_2$, where $C_3$ is an absolute constant. Therefore by the assumptions on $n$ we have

$$|A(\widehat{\mathbf{a}}, \widehat{\mathbf{b}})| \leq C_3 \nu \sqrt{k} + |\mathbb{E}[A(\widehat{\mathbf{a}}, \widehat{\mathbf{b}})]|.$$

By the definition of $\psi_1$-norm, we have

$$\mathbb{E}[A(\widehat{\mathbf{a}}, \widehat{\mathbf{b}})] \leq \|A(\widehat{\mathbf{a}}, \widehat{\mathbf{b}})\|_{\psi_1} \leq C_2 \nu \sqrt{k}.$$

Therefore we have

$$|A(\widehat{\mathbf{a}}, \widehat{\mathbf{b}})| \leq C_4 \nu \sqrt{k}$$

for some absolute constant $C_4$. Now for any $\mathbf{a} \in S^{r-1}$ and $\mathbf{b} \in S^{k-1}$, there exists $\widehat{\mathbf{a}} \in \mathcal{N}_1$ and $\widehat{\mathbf{b}} \in \mathcal{N}_2$ such that

$$\|\mathbf{a} - \widehat{\mathbf{a}}\|_2, \|\mathbf{b} - \widehat{\mathbf{b}}\|_2 \leq 1/4.$$

Therefore

$$A(\mathbf{a}, \mathbf{b}) \leq C_4 \nu \sqrt{k} + I_1 + I_2, \tag{D.3}$$

where

$$I_1 = |A(\mathbf{a}, \mathbf{b}) - A(\mathbf{a}, \widehat{\mathbf{b}})|, \ I_2 = |A(\mathbf{a}, \widehat{\mathbf{b}}) - A(\widehat{\mathbf{a}}, \widehat{\mathbf{b}})|.$$

Let

$$\overline{A} = \sup_{\substack{\mathbf{a} \in S^{r-1} \\ \mathbf{b} \in S^{k-1}}} A(\mathbf{a}, \mathbf{b}).$$

Since $A(\mathbf{a}, \mathbf{b})$ is Lipschitz continuous in $\mathbf{a}$ and $\mathbf{b}$, we have

$$I_1 \leq \|\mathbf{b} - \widehat{\mathbf{b}}\|_2 \overline{A}, \ I_2 \leq \|\mathbf{a} - \widehat{\mathbf{a}}\|_2 \overline{A}.$$

Plugging the two inequalities above into (D.3) gives

$$\overline{A} \leq C_4 \nu \sqrt{k} + \overline{A}/2.$$

Therefore we have $\overline{A} \leq 2C_4 \nu \sqrt{k}$. This completes the proof. $\qquad\square$

## D.8 Proof of Lemma B.8

*Proof of Lemma B.8.* By triangle inequality we have

$$|\sigma(z)| \le |\sigma(z) - \sigma(0)| + |\sigma(0)| \le |z| + |\sigma(0)|.$$

Since $\|z\|_{\psi_2} \le 1$, by the definition of $\psi_2$-norm we have

$$\|\sigma(z)\|_{\psi_2} = \sup_{p\ge 1} p^{-1/2}\{\mathbb{E}[|\sigma(z)|^p]\}^{1/p} \le \sup_{p\ge 1} p^{-1/2}\{\mathbb{E}[(|z| + |\sigma(0)|)^p]\}^{1/p} \le 2[1 + |\sigma(0)|].$$

Similarly, we have

$$|\sigma(z) - \kappa| \le |\sigma(z) - \sigma(0)| + |\sigma(0) - \kappa| \le |z| + |\sigma(0) - \kappa|.$$

By the definition of $\psi_2$-norm, we have

$$\|\sigma(z) - \kappa\|_{\psi_2} \le 2[1 + |\sigma(0) - \kappa|].$$

$\square$

## D.9 Proof of Lemma B.9

*Proof of Lemma B.9.* Let $\mathbf{w}_1, \mathbf{w}_2$ be two distinct unit vectors, and $U_1, U_2$ be jointly Gaussian random variables with $\mathbb{E}(U_1) = \mathbb{E}(U_2) = 0$, $\mathbb{E}(U_1^2) = \mathbb{E}(U_2^2) = 1$ and $\mathbb{E}(U_1 U_2) = \frac{\mathbf{w}^\top(\mathbf{w}_1 - \mathbf{w}_2)}{\|\mathbf{w}_1 - \mathbf{w}_2\|_2} \in [-1, 1]$. Then by the definition of $\phi$, we have

$$
\begin{aligned}
|\phi(\mathbf{w}, \mathbf{w}_1) - \phi(\mathbf{w}, \mathbf{w}_2)| &= \left|\mathbb{E}_{\mathbf{z}\sim N(\mathbf{0},\mathbf{I})}\left[\sigma(\mathbf{w}^\top \mathbf{z})\sigma(\mathbf{w}_1^\top \mathbf{z})\right] - \mathbb{E}_{\mathbf{z}\sim N(\mathbf{0},\mathbf{I})}\left[\sigma(\mathbf{w}^\top \mathbf{z})\sigma(\mathbf{w}_2^{t\top} \mathbf{z})\right]\right| \\
&\le \mathbb{E}_{\mathbf{z}\sim N(\mathbf{0},\mathbf{I})}[|\sigma(\mathbf{w}^\top \mathbf{z})| \cdot |(\mathbf{w}_1 - \mathbf{w}_2)^\top \mathbf{z}|] \\
&\le |\sigma(0)|\|\mathbf{w}_1 - \mathbf{w}_2\|_2 \mathbb{E}(|U_2|) + \|\mathbf{w}_1 - \mathbf{w}_2\|_2 \mathbb{E}(|U_1| \cdot |U_2|) \\
&\le [1 + |\sigma(0)|]\|\mathbf{w}^* - \mathbf{w}^t\|_2 \\
&= L\|\mathbf{w}^* - \mathbf{w}^t\|_2.
\end{aligned}
$$

This completes the proof. $\square$

# E Additional Auxiliary Lemmas

The following lemma is given by [37]

**Lemma E.1** ([37])**.** For two sub-Gaussian random variables $Z_1$ and $Z_2$, $Z_1 \cdot Z_2$ is a sub-exponential random variable with

$$\|Z_1 \cdot Z_2\|_{\psi_1} \le C\|Z_1\|_{\psi_2} \cdot \|Z_2\|_{\psi_2},$$

where $C$ is an absolute constant.

**Lemma E.2.** For any non-zero vectors $\mathbf{u}, \mathbf{v} \in \mathbb{R}^k$, if $\|\mathbf{u} - \mathbf{v}\|_2 \le \rho$, then we have

$$\left\|\frac{\mathbf{u}}{\|\mathbf{u}\|_2} - \frac{\mathbf{v}}{\|\mathbf{v}\|_2}\right\|_2 \le \frac{2\rho}{\|\mathbf{v}\|_2}, \text{ and } \left\langle\frac{\mathbf{u}}{\|\mathbf{u}\|_2}, \frac{\mathbf{v}}{\|\mathbf{v}\|_2}\right\rangle \ge 1 - \frac{2\rho^2}{\|\mathbf{v}\|_2^2}.$$

*Proof of Lemma E.2.* By triangle inequality, we have $\left|\|\mathbf{v}\|_2 - \|\mathbf{u}\|_2\right| \le \|\mathbf{u} - \mathbf{v}\|_2 \le \rho$. Therefore,

$$
\begin{aligned}
\left\|\frac{\mathbf{u}}{\|\mathbf{u}\|_2} - \frac{\mathbf{v}}{\|\mathbf{v}\|_2}\right\|_2 &\le \left\|\frac{\mathbf{u}}{\|\mathbf{u}\|_2} - \frac{\mathbf{u}}{\|\mathbf{v}\|_2}\right\|_2 + \left\|\frac{\mathbf{u}}{\|\mathbf{v}\|_2} - \frac{\mathbf{v}}{\|\mathbf{v}\|_2}\right\|_2 \\
&= \|\mathbf{u}\|_2 \cdot \frac{\left|\|\mathbf{v}\|_2 - \|\mathbf{u}\|_2\right|}{\|\mathbf{u}\|_2\|\mathbf{v}\|_2} + \frac{\|\mathbf{u} - \mathbf{v}\|_2}{\|\mathbf{v}\|_2} \\
&\le \frac{2\rho}{\|\mathbf{v}\|_2}.
\end{aligned}
$$

Moreover, we have

$$-2\left\langle\frac{\mathbf{u}}{\|\mathbf{u}\|_2}, \frac{\mathbf{v}}{\|\mathbf{v}\|_2}\right\rangle + 2 = \left\|\frac{\mathbf{u}}{\|\mathbf{u}\|_2} - \frac{\mathbf{v}}{\|\mathbf{v}\|_2}\right\|_2^2 \le \frac{4\rho^2}{\|\mathbf{v}\|_2^2}.$$

Therefore we have

$$\left\langle \frac{\mathbf{u}}{\|\mathbf{u}\|_2}, \frac{\mathbf{v}}{\|\mathbf{v}\|_2} \right\rangle \geq 1 - \frac{2\rho^2}{\|\mathbf{v}\|_2^2}.$$

This completes the proof. □

The following lemma follows directly from the standard Gaussian tail bound. A similar result is given as Fact B.1 in [41].

**Lemma E.3.** ([41]) Let $\mathbf{w} \in \mathbb{R}^k$ be a fixed vector. For any $t \geq 0$, we have

$$\mathbb{P}_{\mathbf{x} \sim N(\mathbf{0}, \mathbf{I}_k)}\left(|\mathbf{w}^\top \mathbf{x}| \leq \|\mathbf{w}\|_2 \cdot t\right) \geq 1 - 2\exp(-t^2/2).$$

## E.1   Proof of Lemma C.1

*Proof of Lemma C.1.* Let $(\widetilde{X}_1, \widetilde{X}_2)$ be an independent copy of $(X_1, X_2)$. Then by definition we have

$\mathrm{Cov}[f(X_1), g(X_2)] = \mathbb{E}\{[f(X_1) - f(\widetilde{X}_1)] \cdot [g(X_2) - g(\widetilde{X}_2)]\},$

$\mathrm{Cov}[\mathbb{1}(X_1 \leq x_1), \mathbb{1}(X_2 \leq x_2)] = \mathbb{E}\{[\mathbb{1}(X_1 \leq x_1) - \mathbb{1}(\widetilde{X}_1 \leq x_1)] \cdot [\mathbb{1}(X_2 \leq x_2) - \mathbb{1}(\widetilde{X}_2 \leq x_2)]\}.$

For $f(X_1) - f(\widetilde{X}_1)$, by the definition of Lebesgue-Stieltjes integration,

$$f(X_1) - f(\widetilde{X}_1) = \int_{\mathbb{R}} \{\mathbb{1}[x_1 \leq f(X_1)] - \mathbb{1}[x_1 \leq f(\widetilde{X}_1)]\} \mathrm{d}x_1$$

$$= \int_{\mathbb{R}} [\mathbb{1}(x_1 \leq X_1) - \mathbb{1}(x_1 \leq \widetilde{X}_1)] \mathrm{d}f(x_1).$$

Similarly, we have

$$g(X_2) - g(\widetilde{X}_2) = \int_{\mathbb{R}} [\mathbb{1}(x_2 \leq X_2) - \mathbb{1}(x_2 \leq \widetilde{X}_2)] \mathrm{d}g(x_2).$$

Therefore by Fubini's Theorem we have

$$\mathrm{Cov}[f(X_1), g(X_2)] = \int_{\mathbb{R}^2} \mathrm{Cov}[\mathbb{1}(X_1 \leq x_1), \mathbb{1}(X_2 \leq x_2)] \mathrm{d}f(x_1) \mathrm{d}g(x_2).$$

□

*Proof fo Lemma B.10.* For $\{u_t\}_{t \geq 0}$ We have

$$\begin{aligned}
u_t &\leq au_{t-1} + c_1 b^{t-1} + c_2 \\
&\leq a(au_{t-2} + c_1 b^{t-2} + c_2) + c_1 b^{t-1} + c_2 \\
&= a^2 u_{t-2} + c_1(ab^{t-2} + b^{t-1}) + c_2(1 + a) \\
&\leq \cdots \\
&\leq a^t u_0 + c_1(a^{t-1} + a^{t-2}b + \cdots + ab^{t-2} + b^{t-1}) + c_2 \frac{1}{1-a} \\
&\leq a^t u_0 + c_1 t(a \vee b)^{t-1} + c_2 \frac{1}{1-a}.
\end{aligned}$$

This gives the first bound. Similarly, $\{u_t\}_{t \geq 0}$ for We have

$$\begin{aligned}
v_t &\leq av_{t-1} + c_1(t-1)^2 b^{t-1} + c_2 \\
&\leq a(av_{t-2} + c_1(t-2)^2 b^{t-2} + c_2) + c_1(t-1)^2 b^{t-1} + c_2 \\
&= a^2 v_{t-2} + c_1[(t-2)^2 ab^{t-2} + (t-1)^2 b^{t-1}] + c_2(1 + a) \\
&\leq \cdots \\
&\leq a^t v_0 + c_1[0^2 \cdot a^{t-1} + 1^2 \cdot a^{t-2}b + \cdots + (t-2)^2 \cdot ab^{t-2} + (t-1)^2 \cdot b^{t-1}] + c_2 \frac{1}{1-a} \\
&\leq a^t v_0 + c_1 \frac{t(t-1)(2t-1)}{6}(a \vee b)^{t-1} + c_2 \frac{1}{1-a} \\
&\leq a^t v_0 + \frac{c_1}{3} t^3 (a \vee b)^{t-1} + c_2 \frac{1}{1-a}.
\end{aligned}$$

This completes the proof. □

# F  Additional Experiments

In this section we present some additional experimental results on non-Gaussian inputs. Here we consider two types of input distributions: uniform distribution over unit sphere and a transelliptical distribution (the distribution of a Gaussian random vector after an entry-wise monotonic transform $y = x^3$). The experiments are conducted in the setting $k = 15$, $r = 5$ for ReLU and hyperbolic tangent activation functions, where $\mathbf{w}^*$ and $\mathbf{v}^*$ are generated in the same way as described in Section 6. Specifically, Figures 3(a), 3(b) show the results for uniform distribution over unit sphere, while Figures 3(c), 3(d) are for the transelliptical distribution. Moreover, Figures 3(a), 3(c) are for ReLU networks, and the results for hyperbolic tangent networks are given in Figures 3(b), 3(d). From these figures, we can see that although it is not directly covered in our theoretical results, the approximate gradient descent algorithm proposed in our paper is still capable of handling non-Gaussian distributions. In specific, from Figures 3(a), 3(c), we can see that our proposed algorithm is competitive with Double Convotron for symmetric distributions and ReLU activation, which is the specific setting Double Convotron is designed for. Moreover, Figures 3(b), 3(d) clearly show that for hyperbolic tangent activation function, Double Convotron fails to converge, while our approximated gradient descent still converges linearly.

(a) ReLU+Unif. Sphere

(b) tanh+Unif. Sphere

(c) ReLU+Transelliptical

(d) tanh+Transelliptical

Figure 3: Experimental results for different non-Gaussian input distributions. (a) and (b) are for uniform distribution over unit sphere, while (c) and (d) give the results for transelliptical distribiton.