[Reviews · NeurIPS 2019]

Reviewer 1



The authors consider the parameter recovery problem that the data are generated from a teacher network''. The goal is to learn the network from the generated data which are from Gaussian distribution and labeled by the teacher network'' with some white noises. A simple algorithm is proposed to learn the ground-truth parameters of a non-overlapping CNN, which is shown to converge efficiently with high probability with a small sample complexity bound. Due to the good properties of Gaussian inputs, the expectation of each update lies in the span of w^t and w^* (v^t and v^* as well). Also, with some properties of a specific good set'', as long as we haven't yet achieved optimality, the gradient in each step would be large enough and get closer to w^*. Furthermore, once the initial point lies in the good set'', all iterations will still be in that good set to guarantee the high convergence rate. The high-level idea and proof sketch of the paper is clearly written. The authors design a special but simple gradient descent algorithm that has several nice properties to make the steady improvement over iterations happen. However, it would be much better to write down the proof of the claim in line 209 since it is where the assumptions of non-overlapping filter'' and Gaussian data'' are required. Moreover, some conditions used in the proof should be clearly specified. First, the proof indicates that $\eta_w$ and $\eta_v$ are both less than 1 but it is not mentioned in the main text. The c_1 and c_2 in the theorem statement is not enough for showing both $\eta_w$ and $\eta_v$ are less than 1 since C and C' can still be chosen to be large enough such that the inequality does not hold. Secondly, it is not clear how to get line 518 and line 528 from assumptions although I believe that there should be some ways to make the proof work. For the originality and significance of this paper, the authors follow the line of [13] but propose a feasible algorithm that converges linearly. Also, they fix the two-stage convergence guarantee by simply reusing Theorem 4.3 at different time steps. However, this work only applies to non-overlapping and Gaussian input setting, which is far from the cases seen in practice, and it is not clear at all such strong assumptions can be avoided using the approach of this paper. Typos: (a) The P_r in line 101 should be P_k. (b) The third and the last term of step size \alpha (line 152) might have some mistakes. * After author response: I have read the response, but it does not change my opinion. I still feel that the assumptions of Gaussian inputs and non-overlapping filters are very strong, so I really have difficulty in judging the potential impact of this work. On the other hand, I would not be upset if the paper were accepted.

Reviewer 2



Originality: To the best of my knowledge the proposed algorithm as well as the analysis has novel insights/contributions. Quality: The submission is technically sound. The claims are well supported by theory and experiments. Clarity: The paper is very well-written. The introduction and the related work section provide a complete survey as well as comparisons with previous work. The proof section is very well done. Significance: This paper studies an important problem and improves over previous results in several directions. However, the Gaussianity assumption is very restrictive compared to the symmetry assumption in the Convotron paper. Therefore, the algorithm is not likely to have much impact in practice since the analysis heavily relies on the Gaussianity of the input. %%%%%%%%UPDATE%%%%%%%%%% After reading authors feedback, I am happy to increase my rating from 6 to 7. Overall, I vote for accepting this paper.

Reviewer 3



The paper is well written and complete with theoretical proof and experimental results. - The authors present theoretical analysis and experimental proof for modification of the gradient descent algorithm (for non-overlapping CNN case) called approximate gradient descent. The proof desmonstrates that, with high probability, the proposed algorithm with random initialization grants a linear convergence to the ground-truth parameters up to statistical precision. The results are applicable in general non-trivial, monotonic and Lipschitz continuous activation functions including ReLU, Leaky ReLU, Sigmod and Softplus etc. The sample complexity beats existing results in the dependency of the number of hidden nodes and filter size and matches the information-theoretic lower bound for learning one-hidden-layer CNNs with linear activation functions, suggesting that the sample complexity is tight.

[Author Response · NeurIPS 2019]

**To all reviewers:** *on the limitation of problem setting: Gaussian inputs and non-overlapping filters*
Our current result is for Gaussian inputs, and single hidden layer non-overlapping CNNs. However, this is also the case
for most existing results, and even in this very restrictive setting, existing work falls short of tight sample complexity and
generality in terms of activation functions. Therefore the problem we studied in this paper is still largely unsolved before
our work. As is shown in Table 1 in the paper, our result has its unique strength and outperforms the state-of-the-art
results in many aspects. We believe our work is a cornerstone of this line of research, and serves as a foundation towards
more practical results.

**To Reviewer1:**
• *"it would be much better to write down the proof of the claim in line 209... are required. "*
Thanks for your suggestion. We will add the derivation in line 209. It is mainly direct calculation of expectations,
and therefore your suggestion can be easily addressed.

• *"the proof indicates that $\eta_w$ and $\eta_v$ are both less than 1...", "how to get line 518 and line 528 from assumptions",*
*"where and how the assumptions in Theorem 4.3 are used and to explain what the conditions mean."*
Thank you for pointing out these issues. We will revise the statement in Theorem 4.3 to make the conditions
$\eta_w, \eta_v < 1$ clear. We apologize that we missed the assumptions in line 518 and line 528 when merging and
simplifying the conditions on $n$. We will revise the conditions, clarify all the derivations, and discuss their high level
implications in the revision. Here we wish to emphasize that such revision will not affect the overall validity of our
analysis. Except adding several lines of detailed derivations, most parts of our proof will remain unchanged.

• *"Can this algorithm and the analysis be extended to deeper CNN or more layers?"*
Our current analysis is specific to two-layer CNNs, and we have some preliminary idea to extend it to three-layer
CNNs. However, it is not trivial to extend it to deeper CNNs with arbitrary number of layers.

**To Reviewer2:**
• *"plots similar to Figure 1 for non-Gaussian data, or even non-Isotropic Gaussian data."*
Thanks for your suggestion. Here we present experiments with two types of input distributions: uniform distribution
over unit sphere and a transelliptical distribution (the distribution of a Gaussian random vector after an entry-wise
monotonic transform $y = x^3$). The experiments are conducted in the setting $k = 15, r = 5$ for ReLU and hyperbolic
tangent activation functions, where $\mathbf{w}^*$ and $\mathbf{v}^*$ are generated in the same way as Line 253 in our paper. Specifically,
Figures 1(a), 1(b) show the results for uniform distribution over unit sphere, while Figures 1(c), 1(d) are for the
transelliptical distribution. Moreover, Figures 1(a), 1(c) are for ReLU networks, and the results for hyperbolic
tangent networks are given in Figures 1(b), 1(d). From these figures, we can see that although it is not directly
covered in our theoretical results, the approximate gradient descent algorithm proposed in our paper is still capable
of handling non-Gaussian distributions. In specific, from Figures 1(a), 1(c), we can see that our proposed algorithm
is competitive with Double Convotron for symmetric distributions and ReLU activation, which is the specific setting
Double Convotron is designed for. Moreover, Figures 1(b), 1(d) clearly show that for hyperbolic tangent activation
function, Double Convotron fails to converge, while our approximated gradient descent still converges linearly.

(a) ReLU+Unif. Sphere    (b) tanh+Unif. Sphere    (c) ReLU+Transelliptical    (d) tanh+Transelliptical

Figure 1: Experimental results for non-Gaussian input distributions.

**To Reviewer3:**
• *"... works in the ideal setting (as the authors demonstrate during the experimental results, where the unit sphere and*
*standard gaussian vector is used)... "*
We would like to clarify that $\|\mathbf{w}^*\|_2 = 1$ is a reasonable and necessary assumption to ensure generality of our theory.
Note that our theoretical result covers positive homogeneous activation functions like ReLU, leaky ReLU or linear
activation. Without the assumption that $\|\mathbf{w}^*\|_2 = 1$, the true parameters for such networks are *not identifiable*–for
any $c > 0$, the parameters $c \cdot \mathbf{w}^*$ and $c^{-1} \cdot \mathbf{v}^*$ give exactly the same network, and therefore recovering $\mathbf{w}^*$ and $\mathbf{v}^*$ is
impossible. Therefore this assumption essentially specifies a particular set of parameters among a class of equivalent
parameters, and is still a reasonable and necessary assumption even for real setting. We would also like to stress that,
although in our experimental results we generate $\mathbf{v}^*$ as a standard Gaussian vector, this is not essential at all, and
other distributions of $\mathbf{v}^*$, or even manually chosen values of $\mathbf{v}^*$, can always be recovered up to statistical accuracy
by our algorithm. We will add these experimental results in the revision.

[Meta-Review · NeurIPS 2019]

The paper studies the parameter recovery problem in the teacher-student setting for a network with a single hidden layer under the assumption of Gaussian inputs and non-overlapping filters. It proposes a modified training algorithm, shows convergence, and derives sample complexity bounds. All the reviewers appreciated that the contribution is important and the paper is well written. The main concern raised by two reviewers is that the assumptions of Gaussian inputs and non-overlapping filters are very strong, especially considering prior work of Goel et al. has established recovery guarantees for symmetric distributions with a slightly worse sample complexity. Another concern raised by the reviewers during discussions is that the comparisons with previous work in table 1 need clarifications as the previous work of Goel et al. does provide linear convergence in the realizable setting (see Corollary 1 here: https://arxiv.org/pdf/1802.02547.pdf). Even considering these concerns, which the authors should addressed in the final version, the reviewers still feel the paper is above the bar for acceptance.